# Archaeometric studies on rock art at four sites in the northeastern Great Basin of North America

Meinrat O. Andreae [1,2,3]*, Tracey W. Andreae[1]

**1** Max Planck Institute for Chemistry, Mainz, Germany, **2** Department of Geology and Geophysics, King Saud University, Riyadh, Saudi Arabia, **3** Scripps Institution of Oceanography, UCSD, La Jolla, California, United States of America

* m.andreae@mpic.de

## Abstract

Rock art originated some 46,000 years ago and can provide unique insights into the minds of our human ancestors. However, dating of these ancient images, especially of petroglyphs, remains a challenge. In this study, we explore the potential of deriving age estimates from measurements of the areal densities of manganese ($D_{Mn}$) and iron ($D_{Fe}$) in the rock varnish on petroglyphs, based on the concept that the amount of varnish that has regrown on a petroglyph since its creation, relative to the surrounding intact varnish, is a measure of its age. We measured $D_{Mn}$ and $D_{Fe}$ by portable X-ray fluorescence (pXRF) on dated Late Pleistocene and Holocene rock surfaces, from which we derived accumulation rates of Mn and Fe in the rock varnish. The observed rates were comparable to our previous findings on basalt surfaces in North America. We derived age estimates for the rock art at four sites in the northern Great Basin region of North America based on $D_{Mn}$ measurements on the petroglyphs and intact varnish. They suggest that rock art creation in this region began around the Pleistocene/Holocene transition and continued into the Historic Period, encompassing a wide range of styles and motifs. Evidence of reworking of the rock art at various times by Indigenous people speaks of the continued agency of these images through the millennia. Our results are in good agreement with chronologies based on archeological and other archaeometric techniques. While our method remains subject to significant uncertainty with regard to the absolute ages of individual images, it provides the unique opportunity to obtain age estimates for large ensembles of images without the need for destructive sampling.

## Introduction

Rock art is a unique archive of prehistoric human expression, reflecting the societal organizations, subsistence patterns, material culture, and symbolic and religious world of people from the Pleistocene to the recent past. It provides cultural information that is not available from lithic material evidence, such as the existence of a symbolic repertoire and the desire for artistic expression. Rock art has been documented from all continents inhabited by humans [1], as

**Data Availability Statement:** All relevant data are within the manuscript and its Supporting Information files.

**Funding:** The author(s) received no specific funding for this work.

**Competing interests:** The authors have declared that no competing interests exist.

well as from the "Maritime Continent", the islands between Asia and Australia, where the oldest known rock art has been found, with a minimum age of 46 ka BP (BP: calendar years before present, referring to the year 2000 CE) in Sulawesi [2].

In the Americas, the oldest rock art stems from the Terminal Pleistocene/Early Holocene transition. The oldest known example in South America is an anthropomorphic motif from Brazil, dated to around 12–10 ka BP [3]. In the Great Basin of North America, the earliest known rock art dates to the Paleoindian Period, 15 to 9 ka BP, and some of it is possibly even of pre-Clovis age [4, 5]. Cupules and petroglyphs consisting of complex abstract motifs at a site near Lake Winnemucca, Nevada, were carved sometime between 14.8 and 10.3 ka BP [6]. In the southern Great Basin, representational and abstract rock art has been dated to at least 11 ka BP, but possibly extends to as early as ca. 16 ka BP [5, 7, 8]. Hoofprint petroglyphs in South Dakota have been dated by varnish microlamination (VML) to before 14 ka BP [9], and petroglyphs at the Black Rock Creek site in the Wyoming Green River Basin are at least 11 ka old [10, 11]. There is rich evidence that by at least 11 to 10 ka BP, substantial iconographic, stylistic, and technical diversity existed in the rock art of the Americas, including diverse representational and abstract motifs and styles [5, 12, 13].

In this study, we focus on the rock art at four sites in Idaho, Wyoming, and southernmost Montana. This region was chosen because it contains a great variety of rock art ranging in age from Paleoindian to Historic times, and because the availability of previously dated rock art allows a comparison of these dates with our experimental technique [11]. In addition, the presence of dated Holocene lava flows [14] and boulders that were abraded during a massive flood event at 14.5 ka BP [15] in the Snake River Valley provides an opportunity to assess the rate of varnish formation in this area. In a geographic sense, our study region is at the interface between the hydrological Great Basin, the Columbia Plateau, and the Middle Rockies, but we will use the term Great Basin, defined as a cultural and physiographic region by D'Azevedo [16], for simplicity and because of the association of the rock art with the Shoshone people and their ancestors, whose homeland is the Great Basin.

The rock art discussed here consists of petroglyphs, which are produced by removing the dark "rock varnish" coating, present on many rocks in arid or semiarid regions, to form the images. This rock varnish provides a natural canvas, which has been used to create rock art by indigenous people worldwide, from the pre-Neolithic period up to today [e.g., 5, 17–25]. It consists of a matrix of poorly crystallized manganese (Mn) and iron (Fe) oxides and hydroxides (oxyhydroxides), in which clay and other detrital minerals are embedded [26–30]. While the details of the processes by which rock varnish is formed are still the subject of ongoing scientific debate, there is a developing consensus that the Mn and other enriched elements in the varnish matrix, as well as the embedded detrital minerals, are derived from dust deposition and are transformed into the varnish coating by a sequence of dissolution and re-precipitation events [e.g., 31–35]. Further detail and references on this issue can be found in our previous papers [36–38].

Our concept for obtaining age estimates for rock art is based on the fact that once a petroglyph has been created by removing existing varnish, new varnish begins to deposit again on the exposed fresh rock surface. Thus, if the rate of accumulation were known, the amount of the regrown varnish could be used to date the rock art. Using varnish regrowth as an indicator of age has been applied frequently in a qualitative and relative way by visual comparison of the darkness of varnish, e.g., on alluvial fans of different age or on superimposed petroglyphs [19, 39–43]. We have developed a quantitative way to determine the amount of varnish regrowth by measuring the surface density of manganese, the characteristic element of rock varnish, using portable X-ray fluorescence (pXRF) [8, 37, 44–46]. A similar technique had been used previously by Lytle and coworkers [47] and McNeil [48].

We emphasize that these approaches remain experimental and have to be viewed with great caution, because the growth rate of varnish is highly variable and depends on a large number of parameters other than age, including the exposure of the rock surface to dust, erosion by wind and water, the orientation and slope of the rock surface, the hardness, roughness, and texture of the rock underneath, and its initial iron content, as summarized in Andreae and coworkers [8, and references therein]. Some of this variability can be mitigated by normalizing the Mn density on the petroglyph to that of the surrounding intact varnish, which has been subject to the same environmental conditions as the rock art itself [45].

Our earlier work in Arabia has benefitted from the existence of distinct time markers in the Arabian rock art in the form of particular types of scripts, which had been used during specific time periods, and of dated paleoclimatic transitions that are reflected in the animal species depicted in the rock art. Unfortunately, such markers are lacking in North America, and the rate of varnish accumulation has to be estimated based on a variety of assumptions [8]. In this study, we examine the potential of measurements on geologically dated surfaces to derive the varnish accumulation rate. In spite of these uncertainties, however, our previous work in Arabia and North America has shown that the age estimates we obtained were consistent with ages based on the cultural and ecological content of the rock art, and allowed a meaningful ordering of rock images into an age sequence. It has also provided clear evidence for reworking of rock art by subsequent populations, reminding us that rock art has a cultural "life" and "agency", that it is reused and reworked, making the assignment of a single date a questionable concept [49]. This applies especially to our study area, where the present-day Shoshone include rock art and rock art sites in their cultural tradition, and continue to interact with it.

In the present study, we conducted in-situ measurements by pXRF on petroglyphs at four sites in Idaho, Wyoming, and Montana to determine the areal densities of Mn and Fe in the varnish that had re-accumulated on the rock art since its creation. We also made measurements on the intact varnish surrounding the rock art to assess the variability of varnish formation between the different rock surfaces. We complemented these data by measuring the Mn and Fe densities on rock surfaces of known geological age to obtain varnish accumulation rates. From these results, we derived age estimates for the petroglyphs and evaluate them within their archaeological and cultural context. Our study extends significantly the very scarce knowledge base on the rate of rock varnish accumulation in semiarid to mesic regions, provides further validation of a non-destructive dating technique for rock art, and elucidates the history of rock art creation at four important archaeological sites representing the Great Basin Macrotradition.

## Material and methods

### Study region

The four study sites are located in the Snake River Valley of southern Idaho and the Wind River Basin of western Wyoming and adjacent southern Montana (S1 Fig). The vegetation around the sites consists of shrubland and grassland, with sagebrush/juniper or saltbush/prickly-pear and bunchgrass vegetation. In the following sections, we provide a brief description of the geographic and geologic setting and the climate at each of the sites. In addition to the rock art sites, measurements were taken on Holocene lava flow surfaces in Craters of the Moon National Park to obtain rock varnish accumulation rates.

**Celebration Park (10-CN-5) and Map Rock (10-CN-10).** Celebration Park and Map Rock are two of 36 petroglyph sites in the Snake River Valley between Twin Falls and Marsing, Idaho, where almost all of the petroglyphs are engraved into large rounded boulders [15]. These boulders are part of the "Melon Gravel" formation, which consist of basaltic debris

ranging in size from coarse sand to boulders over 5 m in diameter [50]. The boulders are well rounded ("melon-shaped"), as a consequence of being abraded during the transport in the violent flood waters of the Bonneville mega-flood event that went through the Snake River Valley ca. 14.5 ka BP and drained some 4,750 km$^3$ of water from Pleistocene Lake Bonneville.

The petroglyphs at Celebration Park (CP) are in a county park southwest of Boise, Idaho. They are on Melon Gravel boulders, ranging in size between about 1 and 3 m in diameter, on a flood terrace on the northern shore of the Snake River [51, 52]. Since the boulders must have been transported a considerable distance downstream during the flood event, their geological attribution is uncertain, but they originated most likely from the Quaternary basalts that flank the Snake River valley.

Map Rock (MR) is located next to a highway (Map Rock Road), ca. 15 km south of Melba, Idaho [53]. In view of its size and location, it is probably a fallen bolder of the Middle Pleistocene basalts that form the cliffs above the north side of the Snake River [50]. Given its proximity to the river, it is likely, but cannot be proven, that it was also scoured by the Bonneville flood event.

**Legend Rock (48HO4).**   The Legend Rock (LR) site is located in a State Park, 35 km NW of Thermopolis, Wyoming. The rock art is found on a series of panels on a 7 to 10 m high bluff along the north side of Cottonwood Creek, along a distance of some 500 m. Based on the geologic map by Hewett [54], the petroglyphs are engraved in sandstone from the upper (sandstone) member of the Upper Cretaceous Cody Shale formation. Some are on the bluff face well above the present ground level, while others are near ground level or on boulders fallen from the bluff.

**Petroglyph Canyon (24CB601).**   The Petroglyph Canyon (PC) site is just north of the Montana state line, 21 km NW of Lovell, Wyoming. The rock art is found both on the walls of the small canyon itself and on large boulders in the vicinity of the canyon. The petroglyphs are incised in sandstone of the Greybull Sandstone member of the Lower Cretaceous Kootenai Formation (elsewhere referred to as Cloverly Formation) [55].

**Craters of the Moon National Monument.**   To determine rock varnish accumulation rates, we made pXRF measurements on the surfaces of two lava flows in Craters of the Moon National Monument, Idaho: the Blue Dragon Flow dated at 2076 BP (43.45 ºN, 113.53 ºW, 1750 m a.s.l.) and the North Crater Flow dated at ca. 2400 BP (43.46 ºN, 113.56 ºW, 1790 m a.s.l.) [14, 56]. Both flows consist of trachybasalts [57]. The measurements were taken on smooth pahoehoe flow surfaces that showed no visual evidence of erosion or alteration.

## Climate and paleoclimate

The Boise/Nampa region has a steppe climate, with a Köppen-Geiger classification of Bsk (cold semi-arid). The annual rainfall is 305 mm, with a maximum in December and a minimum in July. The average temperature is -0.2 ºC in January and 27.1 ºC in July (https://en.climate-data.org/north-america/united-states-of-america/idaho-5/ last accessed 23 April 2021).

The Thermopolis area has a cold temperate climate, with a Köppen-Geiger classification of Dfa (Cold-continental, hot summer humid). In spite of the difference in classification, the annual precipitation is similar to the Boise/Nampa area with total of 319 mm, peaking in May with summer and winter minima. The hottest month is July, with an average temperature of 24.3 ºC and the coldest is January, with an average of -7.7 ºC. The Lovell region is slightly drier at 300 mm annual precipitation, placing it in the Köppen-Geiger classification of Bsk. The July and January average temperatures are 24.8 ºC and -4.5 ºC, respectively (https://en.climate-data.org/north-america/united-states-of-america/wyoming-925/ last accessed 23 April 2021).

At the Last Glacial Maximum (LGM, ca. 20 ka BP) the Snake River Plain and western Wyoming experienced a dry and cold climate, with dominant dry easterly winds and temperatures about 4–5 ºC lower than today [58]. In the Early Holocene (11.5 to 9 ka BP) temperatures warmed, but dry conditions kept supporting steppe climate and vegetation in the region [59, 60]. Peak xeric (warm and dry) conditions have been proposed to have occurred around 7–8 ka BP, followed by slightly cooler and moister conditions similar to the present [59]. Overall, dry conditions, interrupted by occasional moister episodes, dominated in the region throughout the Holocene [61], providing an environment suitable to rock varnish deposition.

## Prehistory

Humans arrived in the northern Great Basin around 16 ka BP, with the archaeologic evidence suggesting nomadic hunting/gathering lifeways [4, and references therein, 62–66]. The ethnographic identity of the first inhabitants of the region is unclear, and even the existence, timing, and directionality of the Numic (Shoshone) expansion across the Great Basin continues to be debated [67]. While Bettinger and Baumhoff [68] proposed an arrival of Numic speakers as late as 500–700 years ago, Husted and Edgar [69] argued that "the Eastern Shoshone and their ancestors have been in Wyoming for 10,000 years". In the Late Prehistoric Period, beginning about 1500 BP, Siouan (Crow) and Algonquian (Arapaho) speaking people moved into parts of the region [17]. Regardless of their ethnographic identity, nomadic or semi-nomadic hunting-gathering lifeways dominated in the region through the Protohistoric Period (ca. 1700 to 1805 CE, [66, 70]) and into the Historic Period, until the forcible removal and resettlement of the native tribes into reservations in the second half of the 19[th] century CE.

## Rock art styles and chronology

The rock art styles in the study region are part of the Uto-Aztecan Rock Art Macrotradition or Great Basin Macrotradition [11, 71]. This macrotradition encompasses a large number of traditions and styles, which were in used over various lengths of time from the Paleoindian to the Historic Periods. For example, the Great-Basin Carved Abstract (GBCAS) style, consisting of deeply carved dense geometric patterns, was dated to the Terminal Pleistocene/Early Holocene (TP/EH) period between 13 and 9 ka BP [4]. The sites investigated in our study include rock art from several traditions or styles within the Great Basin Macrotradition, specifically the Great Basin Pecked (GBP), the Legend Rock Outline Complex (LROC), Dinwoody Anthropomorphs (DA), and En-Toto Pecked (ETP). In addition, there are some images from the Historic Period. Given the complex and sometimes inconsistent terminology and classification of the rock art in our study area, we are including a brief outline of the classification used here, which is mostly based on recommendations from a 2016 workshop [71].

In the following discussion, we use a terminology where "element" means a single form or design unit, often used synonymously with "petroglyph" or "image". A "motif" is an element that is often used within a given corpus and is related to a particular style, e.g., an atlatl or a bighorn sheep.

**Great Basin Pecked style.** The Great Basin Pecked (GBP) style originated in the southern Great Basin (GB) around 11 ka BP, possibly even as early as 13 ka BP, and continued into the 20[th] century [72]. It is thought to have arrived in the northeastern GB considerably later than in the southwestern GB core area, but the precise timing has so far been uncertain because of the lack of dated rock art representing this style in the northeastern GB. As we will show below, our measurements suggest that this style has been present in the Snake River Valley since at least 10 ka BP. The GBP style encompasses simple geometric designs as well as stylized representational motifs, commonly bighorn sheep and anthropomorphs.

**Legend Rock Outline Complex.** Originally included with the Dinwoody Tradition by Francis and Loendorf [17], the LROC complex is the designation agreed upon at the 2016 Dinwoody Dissected workshop for the highly realistic depictions of large artiodactyls (e.g. elk, bison, deer) rendered by first pecking the outline of the torso and then sometimes filling the interior (or portions thereof) with pecked interior lines, or even solid pecking. They are sometimes associated with simple outline-pecked humans with round heads and out-stretched arms. Tratebas [73] refers to the LROC as the Old Outline Pecked Animals style. The LROC is typified by Panel 35 at Legend Rock, and likely at site 48FR99 [71]. This definition includes some of the motifs previously included in Early Hunting and is identical to the Old Outline Pecked Animals style. The LROC is thought to be of Paleoindian to Early Archaic age, consistent with the heavy varnish on these elements. For LROC elements on Panel 35, VML, $^{14}$C accelerator mass spectrometry (AMS), and cation ratio (CR) dating indicated ages from 10.6 to 6.2 ka BP. AMS dating on LR Panel 48 yielded ages around 5.9 ka, and other elements on this panel are likely of similar age [74].

**Dinwoody Tradition.** The Dinwoody Tradition, as defined by Francis and Loendorf [17] [see also 71], is geographically restricted to the Bighorn and Wind River Basins of Wyoming. Ethnographic evidence suggests that Dinwoody Tradition petroglyphs were created by Numic speaking peoples, particularly Shoshone shamans and shaman initiates, to obtain supernatural power for the purposes of curing [11, 17, 75]. Most Dinwoody petroglyphs are produced by pecking, but abrasion and incision also occur, sometimes in combination.

The most recognizable Dinwoody motifs are highly distinctive, often near life-size, pecked anthropomorphic figures, designated here as Dinwoody Anthropomorphs (DA). These are characterized by complex, surreal, non-naturalistic human and therianthropic (human-animal) images, generally with some type of headdress and often bizarre orientations of arms and legs, and disproportionately sized hands and feet. There is a tremendous variety of the ways heads, eyes, limbs, hands, headgear, animal appendages, etc. are represented [75, 76]. Many DAs exhibit patterns of interior lines in the torso and thus were originally termed the Interior Line style by Gebhard and Cahn [77] based upon their early investigations at Dinwoody Lake on the Wind River Indian Reservation. However, many other DAs have fully pecked or stipple pecked torsos or smaller anthropomorphic images within the torsos [11, 17]. As a result, we use the designation DA to refer this general group of anthropomorphic images. Similar types of anthropomorphic figures, many with patterned bodies, occur in several regions across the Great Basin from California to Wyoming [76, 78, 79] and are thought to be associated with Uto-Aztecan speakers.

Until recently, the Dinwoody Tradition was thought to encompass the Early Hunting (EH), ETP, and Dinwoody Interior Line and Fully-Pecked styles, but consensus at the 2016 Dinwoody Dissected workshop suggested a restructuring, where these styles are considered as separate entities [71]. The workshop also suggested not to use the EH terminology in the Bighorn and Wind River Basins [11].

The DAs are often associated with small, solidly pecked zoomorphs, which include artiodactyls (mountain sheep, bison, elk, deer, etc.), canids, bears, birds, reptiles, amphibians, and insects [11, 80]. These zoomorphs are most often rendered both in fully pecked, and less frequently, outline pecked manners and are generally more stylized than LROC animals. Their association on the same panels with DAs suggest similar ages. Many of these zoomorphic figures have previously been referred to as EH [77] or ETP [81].

Previous age estimates for Dinwoody Tradition petroglyphs range from 5.4 ka (CR estimate on a fully pecked canid on LR Panel 74) to 225 a BP (AMS age on a fully pecked rabbit on LR Panel 78) [74], and most dates for the DAs range between 1 and 3 ka BP [11]. However, many

DA images have been reworked in order to restore their power or acquire power from them [82], which would result in younger apparent ages.

**En-toto Pecked.** The type locality for the En-toto Pecked (ETP) style is Petroglyph Canyon (PC; 24CB601) [81]. Anthropomorphs are the most common motif, often shown in groups. They are frequently depicted with head-gear, fingers, genitals, and occasionally carrying bows. The animals represented in the ETP style are mostly ungulate species, such as pronghorn, bison, and sheep, but do not include elk. Other animal motifs include canids and birds ("Thunderbirds"). In many instances they are so stylized that the species represented cannot be identified. Abstract motifs are rare. The ETP style is relatively recent, likely ranging from about 2600±350 BP for a quadruped at PC to <1000 BP for a Thunderbird at the same locality.

**Historic Period.** Following the arrival of Euro-Americans, horses and trade goods reached the Indigenous tribes in late 17th century. As a result, rock art from the Historic Period includes horses, rifles, houses, etc., as well as names and dates inscribed in the rock faces.

## Methods

### Portable X-ray fluorescence spectrometry

The pXRF measurements were conducted using a Niton XL3 pXRF (Thermo Fisher Scientific) in the "Mining" mode, which refers to the internal software setting appropriate for geochemical samples. The instrument is equipped with an X-ray source with an energy of 50 keV and a silver anode, and has a spot size of 8 mm in diameter. During the measurement cycle, a sequence of filters are placed in the path of the X-rays and the anode voltage is adjusted to be optimal for elements with different atomic number, Z. The filter steps and integration periods over which measurements were made with each filter setting were: "Standard": 25 s, "Low-Z": 15 s, "High-Z": 20 s, and "Light Elements": 25 s. The measurement depth is dependent on the energy of the excitation and fluorescence photons, as well as on the composition (atomic number) of the analyte. The software in the instrument takes these factors into account when calculating the results. For quality control, the reference materials TILL-4 and FeMnOx-1 [GeoReM database, version 25; http://georem.mpch-mainz.gwdg.de; 83] were measured before and after each XRF measurement sequence. For the elements considered here, the measurement depth is of the order of a few tens to hundreds of microns. Further detail can be found in our previous publications [37, 45].

A total of 461 measurements were made, 245 on intact varnish surfaces, 209 on petroglyphs, and the rest for ancillary purposes, e.g., on freshly exposed bare rock substrates. Multiple measurements were made on several spots inside and adjacent to petroglyphs, to obtain a representative average value for each petroglyph and its surrounding intact varnish. Depending on the size and complexity of a given rock art element, two to six such spots (average: three) were measured on each element. The spots were visually selected to represent as much as possible homogeneous and undisturbed varnish. The measurement points on the adjacent intact varnish were chosen to be as close as possible to the petroglyph measurement spots, and to be as similar as possible in surface characteristics. Care was taken during the measurements to avoid spots with residual varnish that was left unpecked by the artist, but sometimes very small spots may not have been visible. In cases where no spots of 8 mm diameter without residual varnish could be found, the "small spot" mode (3 mm diameter) of the pXRF was selected.

The pXRF measurements, reported by the instrument as mass percentages of elements in the rock sample, were converted into areal density values, $D_{Mn}$, in units of µg cm$^{-2}$ using the calibration curve from Macholdt and coworkers [84]. To correct for the underlying host rock element contribution, the Mn concentration on a nearby freshly exposed rock surface was subtracted from that measured on the varnished surfaces. The areal density of Fe ($D_{Fe}$) was

calculated using the Mn calibration values and the Mn/Fe sensitivity ratio, and is thus subject to a greater uncertainty (estimated at about 20%). Since $D_{Mn}$ and $D_{Fe}$ vary substantially due to different growth and erosion conditions even within each rock art panel location, we also calculated the ratio of the measurements on the petroglyph surfaces to that on immediately adjacent intact varnish. This provides a normalized measure, called $N_{Mn}$ and $N_{Fe}$ (in %), which expresses the degree of re-varnishing on the petroglyph surface relative to the surrounding intact varnish. The measurement and data reduction techniques used were identical to those in Macholdt et al. [45] and are described in more detail there. Photographs of all petroglyph measurement locations are shown in the Supplement.

Permission for the measurements at Celebration Park and Map Rock was obtained from the Canyon County Parks Program Manager and permission for the work at Legend Rock from the Wyoming State Parks Superintendent. Permission for the measurements in Craters of the Moon National Monument was provided by the Monument Superintendent. No permit was required for the collection of small rock samples and non-invasive measurements by pXRF on public lands. No samples of cultural heritage material were taken.

### Data analysis

Regression calculations were made using bivariate regression, which takes into account error in both the dependent and independent variables, using the Williamson-York Iterative bivariate fit algorithm [85].

## Results and discussion

### Areal density of manganese and iron in the rock varnish

Summary statistics from the results of our pXRF measurements on rock varnish are presented in Tables 1 and 2. The absolute surface densities, $D_{Mn}$ and $D_{Fe}$ (Table 1), on intact varnish show a pronounced difference between the varnishes on the Pleistocene basalts (CP and MR) and on sandstone (LR and PC), with the Mn densities on basalt being two to four times higher than on sandstone. Although regional factors can of course not be excluded, it is likely that this

**Table 1.** Mn and Fe areal densities and Mn/Fe ratios from pXRF measurements on intact rock varnish and petroglyphs at the study sites [Avg.: Arithmetic average, S.D.: Standard deviation].

| Site | N | $D_{Mn}$ | | $D_{Fe}$ | | Mn/Fe | |
|---|---|---|---|---|---|---|---|
| | | [μg cm$^{-2}$] | | [μg cm$^{-2}$] | | | |
| | | Avg. | S.D. | Avg. | S.D. | Avg. | S.D. |
| Intact varnish | | | | | | | |
| Celebration Park | 69 | 430 | 190 | - - - | - - - | - - - | - - - |
| Map Rock | 27 | 200 | 160 | - - - | - - - | - - - | - - - |
| Old flow surface at MR | 5 | 420 | 100 | 580 | 240 | 0.81 | 0.32 |
| Legend Rock | 66 | 81 | 41 | 240 | 140 | 0.52[a] | 0.29[a] |
| Petroglyph Canyon | 35 | 76 | 49 | 410 | 370 | 0.41[a] | 0.16[a] |
| Craters of the Moon | 25 | 52 | 25 | 200 | 170 | 0.25 | 0.24 |
| Petroglyphs | | | | | | | |
| Celebration Park | 73 | 135 | 97 | - - - | - - - | - - - | - - - |
| Map Rock | 29 | 25 | 22 | - - - | - - - | - - - | - - - |
| Legend Rock | 68 | 32 | 23 | 180 | 120 | 0.32[a] | 0.29[a] |
| Petroglyph Canyon | 34 | 13 | 10 | 140 | 60 | 0.14[a] | 0.07[a] |

[a]) outliers with Mn/Fe<0.05 removed

**Table 2. Normalized Mn and Fe areal densities from pXRF measurements on petroglyphs at the study sites [Avg.: Arithmetic average, S.D.: Standard deviation].**

| Site | N | $N_{Mn}$ [%] | | $N_{Fe}$ [%] | |
|---|---|---|---|---|---|
| | | Avg. | S.D. | Avg. | S.D. |
| Map Rock | 9 | 9.5 | 9.2 | - - - | - - - |
| Celebration Park | 21 | 32 | 19 | - - - | - - - |
| Legend Rock | 22 | 34 | 22 | 81 | 32 |
| Petroglyph Canyon | 10 | 15 | 9 | 48 | 21 |

difference is due to the greater weathering resistance of the basalts, which allows varnish to accumulate for a longer time before being removed again by weathering. This is supported by our previous results on these two different substrates: on Pleistocene basalts in the Mojave Desert, we measured an average $D_{Mn}$ of 530±290 µg cm$^{-2}$ [8], whereas the average $D_{Mn}$ values from our five sites in Saudi Arabia, all of which are on sandstone, range from 105 to 210 µg cm$^{-2}$ [37, 44–46]. The overall lower $D_{Mn}$ at our current study sites might be related to the wetter and colder climate at these more northerly locations being less favorable for varnish formation than at our previous, more southerly and drier sites. Precipitation at the present sites is about twice as high (around 300 mm a$^{-1}$) than at our previous ones (around 150 mm a$^{-1}$), and thus likely above the optimum for varnish formation proposed by Dorn and Oberlander [86]. The few available measurements of $D_{Mn}$ in North America by other authors show a similar trend with low values on sandstone (25–90 µg cm$^{-2}$) at more mesic sites in Utah (ca. 250 mm a$^{-1}$) [48] and higher values (90–220 µg cm$^{-2}$) on clasts of unknown lithology at sites in the Mojave Desert [41]. The lowest $D_{Mn}$ values are from the late Holocene lava flows in Craters of the Moon NM, consistent with their young age of about 2.2 ka.

The average $D_{Mn}$ values on the petroglyphs at the four sites vary between 9.5 and 34% relative to those on the adjacent intact varnish, reflecting the shorter length of time that they were exposed to varnish accumulation.

Because of the high and variable iron concentration in the underlying basalt, we were not able to obtain valid $D_{Fe}$ data for the varnishes on basalt, with the exception of Craters of the Moon NM and some measurements on a pahoehoe lava flow surface on a ledge above MR, where we measured an average value of 580±240 µg cm$^{-2}$, similar to our value of 630±320 µg cm$^{-2}$ on the Mojave Desert basalts. The much lower Fe concentration in the sandstones allowed reliable $D_{Fe}$ measurements, which gave values comparable to those we had obtained at our Arabian sandstone sites.

The Mn/Fe ratio of rock varnishes is a useful parameter, as it allows discriminating between true manganese-rich rock varnishes and iron-oxide coatings or fracture fills, which typically have Mn/Fe ratios much below 0.1 [46, 86]. Because of the difficulties with the subtraction of the Fe values of the basalt substrate, as discussed above, we were able to obtain Mn/Fe ratios only from the varnish on the sandstone, on the pahoehoe flow surface at MR, and on the lava flows at Craters of the Moon NM. The value on this flow surface is 0.81±0.32, close to our value of 0.86±0.63 from the Mojave Desert. The average Mn/Fe ratios of both the intact varnish and the petroglyphs on the sandstones (Table 1) are near the lower end of the range of 0.22 to 1.30 measured previously in North America and the Middle East [8, 35, 37, 44–46, 87–91]. The observed Mn/Fe ratios of <1 in the varnish at our sites are consistent with a formation during the Holocene or late Pleistocene, as microstratigraphic analyses by Liu and coworkers [92–94] had shown that the Holocene was characterized by a yellow layer (in

microscope thin sections) with Mn/Fe <1 formed under arid conditions, whereas varnish from the last glacial period contained black layers with Mn/Fe up to ~4 reflecting wetter periods.

While the average Mn/Fe ratios are within the range of true rock varnish, some of the individual measurements fall below the ratio of 0.05 that is characteristic of true Mn-rich varnish (Fig 1). At very low $D_{Mn}$ values, this may simply be the result of numerical uncertainties close to the detection limit or to the presence of an Fe-oxyhydroxide enrichment near the rock surface. A reddish-brown Fe-oxyhydroxide layer is often clearly visible in the porous weathering rinds of the sandstones we studied in Arabia, penetrating as much as several mm into the substrate [for images see 45]. Furthermore, a high-Fe coating can form very rapidly, as seen for example on a surface created by uranium prospecting in the 1950s, which already had a $D_{Fe}$ of 250 μg cm$^{-2}$, but no significant Mn. Similar rapid formation of Fe-coatings has been previously found at our Arabian sandstone sites [37, 45, 46].

Especially at PC, however, there is one subset of the data that clearly differs by having very high $D_{Fe}$ and very low Mn/Fe (Fig 1B), suggesting that they represent a rock coating formed by a process other than that which yields true Mn-rich rock varnish. Such reddish Fe-rich coatings can form either in hyperarid environments or at locations where the physicochemical conditions (Eh, pH) are not conducive to rock varnish formation [86]. Consequently, they are not likely to yield accurate age estimates.

## Normalized areal density of manganese and iron on the petroglyph surfaces

We define the normalized Mn areal density, $N_{Mn}$, as the areal density of Mn on a petroglyph surface divided by that on the adjacent intact rock varnish surface, expressed in percent [37, 45, 84]. This value can be considered as the regrowth percentage of the varnish on a petroglyph following its creation by removal of the preexisting varnish. This normalization accounts, at least in part, for the effects of the numerous variables other than exposure time, which affect the rate of varnish accumulation, such as the inclination of the rock surface, microclimate, water runoff, substrate characteristics, etc.

The average $N_{Mn}$ and $N_{Fe}$ data from our sites are given in Table 2. Again, no $N_{Fe}$ data could be obtained for the measurements on basalt because of the variability of Fe in the basalt substrate. Similar to our previous work, we find little or no correlation between $N_{Mn}$ and $N_{Fe}$, and no evidence for a relationship between $N_{Fe}$ and age. We will thus focus our subsequent discussion of varnish growth in the petroglyphs on $N_{Mn}$.

## Variability of the Mn areal densities at various scales

The great variability of varnish areal density is evident even to the naked eye on any varnish covered outcrop surface. This variability exists at all scales, from the microscopic to the landscape scales. The extreme variability at the microscale, ranging from uncoated mineral grains to thick varnish in microbasins [95–97] is averaged over by the spot size (8 mm diameter) of the pXRF measurement. Within a given site, the variability at scales of decimeters to hundreds of meters is reflected by the standard deviations of the measurements on intact varnish, which are typically on the order of 50% (Table 1), and by ranges of one to two orders of magnitude between individual measurements (Fig 1). At continental scales, on the other hand, we find lower variability: The site-average $D_{Mn}$ of all our data from North America and Arabia fall within a range of about three each for sandstone (76 to 210 g cm$^{-2}$) [37, 44–46] and for basalt (200 to 750 μg cm$^{-2}$) [Table 1; 8].

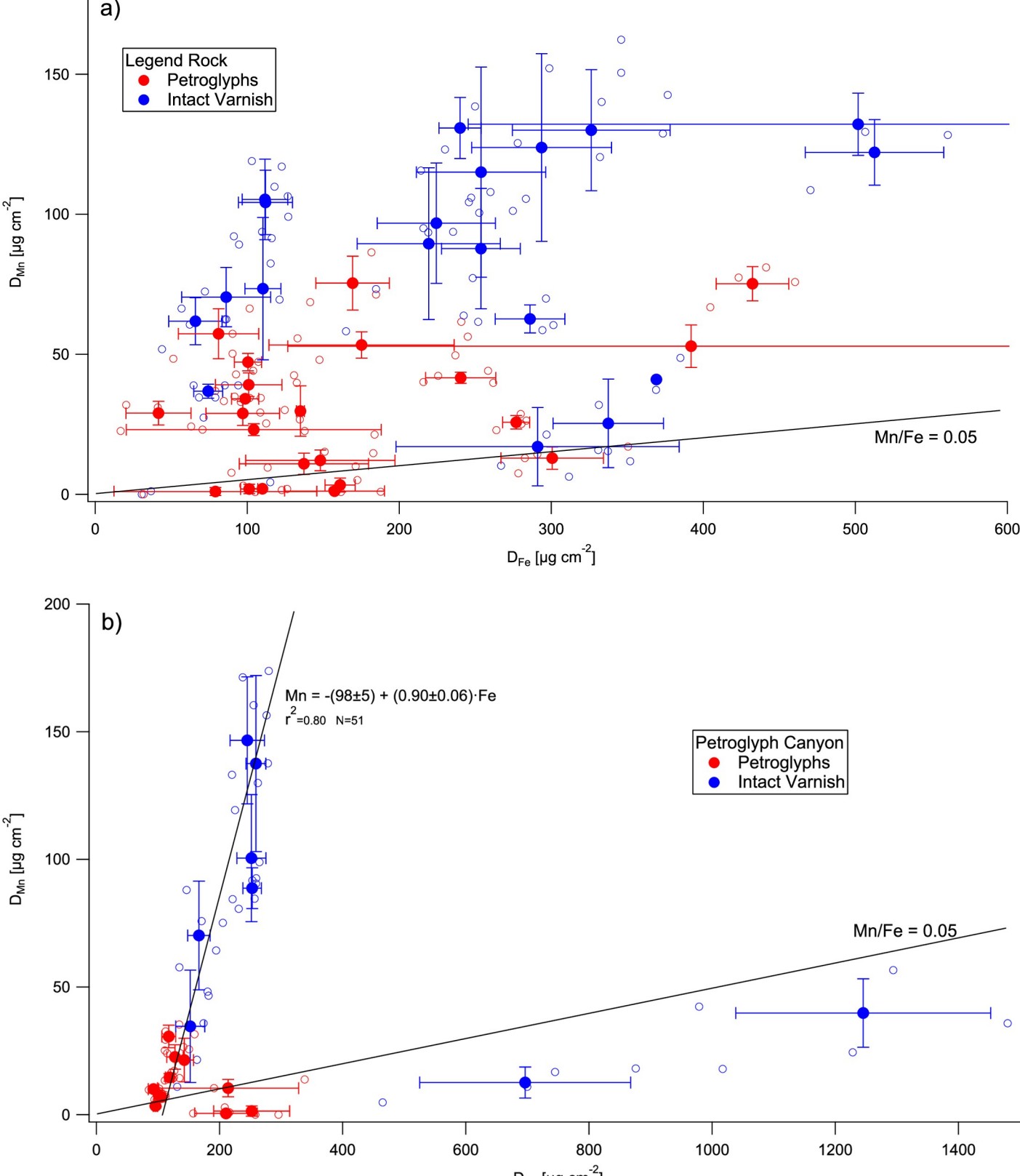

**Fig 1.** Areal density of Mn vs. areal density of Fe on the rock varnishes at the (a) Legend Rock and (b) Petroglyph Canyon sites. Individual measurements are shown as open symbols, rock art element averages as filled symbols. Error bars are standard deviations.

In the context of using Mn density measurements for obtaining rock art age estimates, the most important variability is that at the cm to m scale, i.e., within rock art elements and panels. While this variability is often emphasized in the literature and cited as an obstacle to dating, it is rarely expressed in quantitative terms. To address this issue, we make multiple measurements within and adjacent to a given petroglyph, typically three each. We can then quantify the within-element variability by the coefficients of variation (CV), i.e., the ratio of the standard deviation over the mean (expressed as percentage), of the areal density measurements on the various types of varnished surfaces. We represent the uncertainty of the mean for each set of measurements on a particular element or surrounding varnish by the standard error (SE), which takes into account the number of replicate measurements.

The average CV for the $D_{Mn}$ on intact varnish surrounding a particular petroglyph is 24.5% (N = 45; range 5.3–83%) and the corresponding average relative SE is 13.0% (range 3.4–38). The variability on the rock art elements is almost identical, with an average CV of 23.9% (N = 55; range 2.6–75%) and an average relative SE of 13.9% (range 1.5–53%) (four values with $D_{Mn}$ near the detection limit were excluded, since they had very large CV (>100%) in spite of small absolute errors). These variability statistics are very similar to the results from our previous studies and appear to be typical of varnish variability at the rock art scale. They indicate that the uncertainty due to measurement error and within-element variability is relatively modest, about 14%.

The CVs of the normalized densities, $N_{Mn}$, are identical to those of the absolute densities, since the measurements on each petroglyph were always normalized by dividing them by the average of the measurements on the corresponding intact varnish. The mean standard error of the $N_{Mn}$ data was estimated by error propagation from the CVs of the petroglyph and intact varnish measurements and the corresponding number of replicates, and was found to be 20.3% (range 6.2–61%). The absolute SE values for the measurements on the individual rock art elements are given in Tables 3 to 6.

## Absolute and normalized Mn accumulation rates

A key prerequisite for obtaining rock art age estimates from Mn densities is knowledge of the accumulation rate of the varnish, here represented by its Mn accumulation rate. This rate, $R_{Mn}$ (in units of $\mu g\, cm^{-2}\, ka^{-1}$), is defined as the ratio of the Mn areal density, $D_{Mn}$, on a rock surface to the known or estimated exposure age of this surface. In the simplest case, the age of a petroglyph can then be obtained by dividing its $D_{Mn}$ by $R_{Mn}$. Note, however, that $R_{Mn}$ is the average rate of Mn accumulation over its entire exposure age, ignoring any variations of the true instantaneous accumulation rate that are likely to have occurred. In a previous study, we have shown that at very long time scales, varnish accumulation slows down as the varnish density on the petroglyph surface approaches that on the surrounding intact varnish, and varnish deposition is balanced by its removal by weathering and erosion. This issue becomes important at time scales that approach the effective age of the rock surface, also referred to as the "taphonomic limit", which in our previous studies we found to be reached at ages around 10 ka on sandstone [37, 45]. On basalt, it is likely to be significantly higher because of the greater weathering resistance of this rock type [8]. On the other hand, the very existence of the rock varnish, silica skins, and/or oxalate deposits at the rock surface may increase its weathering resistance through a process called "case hardening" [9, 25, 98] resulting in a greater effective age of varnished sandstone than bare rock of the same composition.

To address these issues and obtain robust location-specific accumulation rates would require availability of multiple rock or petroglyph surfaces of known and diverse ages. In Arabia, we benefited from the inscriptions of known age or dated ecological transitions reflected

**Table 3. Mn absolute ($D_{Mn}$) and normalized areal densities ($N_{Mn}$) on the rock art elements at Celebration Park and the corresponding age estimates based on $N_{Mn}$ [Avg.: Arithmetic average, S.D.: Standard deviation, S.E.: Standard error].**

| Element | Motif | Style | N | $D_{Mn}$ | | | $N_{Mn}$ | | Age estimate | |
|---|---|---|---|---|---|---|---|---|---|---|
| | | | | Avg. | S.D. | S.E. | Avg. | S.E. | Avg. | Error |
| | | | | [µg cm$^{-2}$] | | | [%] | | [ka BP] | |
| CP2120 | Arch & dots | GBCA | 4 | 60 | 9 | 4 | 9% | 1% | 1.4 | 0.4 |
| CP2108 | Rake | GBRA | 3 | 22 | 10 | 6 | 12% | 3% | 1.7 | 0.5 |
| CP2208 | Complex pattern | GBCA | 3 | 59 | 26 | 15 | 13% | 4% | 1.9 | 0.6 |
| CP2166 | Lizard | GBR | 3 | 62 | 11 | 6 | 15% | 2% | 2.1 | 0.6 |
| CP2185 | Sun | GBCA | 3 | 67 | 10 | 6 | 17% | 2% | 2.4 | 0.7 |
| CP2114 | Arch & dot | GBCA | 3 | 43 | 18 | 10 | 17% | 5% | 2.5 | 0.7 |
| CP2215 | Complex pattern | GBCA | 6 | 60 | 11 | 5 | 18% | 3% | 2.5 | 0.8 |
| CP2179 | Triangle & dots | GBRA | 2 | 50 | 19 | 13 | 20% | 7% | 2.9 | 0.9 |
| CP2172 | Complex pattern | GBCA | 4 | 72 | 10 | 5 | 21% | 8% | 3.1 | 0.9 |
| CP2191 | Wavy line | GBCA | 4 | 124 | 55 | 28 | 24% | 6% | 3.5 | 1.0 |
| CP2102 | Animal? | GBR? | 3 | 90 | 32 | 19 | 27% | 7% | 4.0 | 1.2 |
| CP2235 | Star pattern | GBCA | 5 | 220 | 72 | 32 | 35% | 6% | 5.0 | 1.2 |
| CP2198 | Anthrop., body, reworked | GBR | 3 | 111 | 22 | 13 | 28% | 4% | 4.0 | 1.5 |
| CP2201 | Anthrop., arm, original | GBR | 3 | 142 | 37 | 21 | 36% | 7% | 5.2 | 1.5 |
| CP2160 | Rake | GBRA | 3 | 140 | 52 | 30 | 42% | 25% | 6.1 | 1.8 |
| CP2140 | Arch & lines, reworked | GBMA | 3 | 133 | 5 | 3 | 42% | 9% | 6.2 | 1.8 |
| CP2127 | Arch & dot | GBCA | 3 | 290 | 64 | 37 | 46% | 5% | 6.6 | 2.0 |
| CP2226 | Circle chain | GBCA | 5 | 170 | 39 | 17 | 51% | 9% | 7.4 | 2.2 |
| CP2133 | Line pattern | GBRA | 3 | 350 | 83 | 48 | 53% | 6% | 7.7 | 2.3 |
| CP2152 | Concentric circles pattern | GBCA | 4 | 280 | 89 | 44 | 66% | 8% | 9.6 | 2.9 |
| CP2146 | Line pattern, no reworking | GBRA | 3 | 240 | 52 | 30 | 77% | 11% | 11.2 | 3.4 |

in the rock art motifs, which are unfortunately not available at our present sites. The only unambiguously dateable surfaces here are the "Melon Boulders" at Celebration Park, which were formed by fluvial erosion during the Bonneville flood event at 14.5 ka BP [15] and the lava flow surfaces in Craters of the Moon National Monument. From the Melon Boulders, using the average $D_{Mn}$ of 410±140 µg cm$^{-2}$ on the intact varnish surrounding the 20 rock art elements measured at CP, we obtain an $R_{Mn}$ of 28.2±9.9 µg cm$^{-2}$ ka$^{-1}$. This value is in the range of our previous $R_{Mn}$ from basalt flows of similar age in California (15 to 21 µg cm$^{-2}$ ka$^{-1}$) [8] and the estimate of 30 µg cm$^{-2}$ ka$^{-1}$, which we derived from the measurements of Reneau [41] on Holocene surfaces in the Mojave Desert [37]. It is somewhat higher than our $R_{Mn}$ values on sandstone-based mid- to late Holocene petroglyph surfaces from the Hima (13.4 µg cm$^{-2}$ ka$^{-1}$), Ha'il (17 µg cm$^{-2}$ ka$^{-1}$) and al-Jawf (14 µg cm$^{-2}$ ka$^{-1}$) regions in Saudi Arabia [37, 44, 45].

To obtain Mn accumulation rates from more recent surfaces and to test our assumption of an approximately constant accumulation rate, we conducted measurements on two trachyba-salt pahoehoe lava flows in Craters of the Moon National Monument in the Snake River Plain of southern Idaho. On the Blue Dragon flow (2076 BP), we obtained an average $D_{Mn}$ of 47.4 ±25.4 µg cm$^{-2}$ (N = 19), yielding an $R_{Mn}$ of 22.8±12.3 µg cm$^{-2}$ ka$^{-1}$. The North Crater flow (ca. 2.4 ka BP) had an average $D_{Mn}$ of 68.8±17.3 µg cm$^{-2}$ (N = 7), corresponding to an $R_{Mn}$ of 28.7 ±7.2 µg cm$^{-2}$ ka$^{-1}$. The overall $R_{Mn}$ of 25.6±12.4 µg cm$^{-2}$ ka$^{-1}$ from these two ~2000-year-old flows is in excellent agreement with the accumulation rate on the 14.5 ka Melon Gravels, supporting our assumption of a near-linear accumulation rate over the time spans of concern for our study.

A somewhat less robust estimate of the Mn accumulation rate was obtained at Map Rock. This boulder does not have the highly rounded shape of the Melon Gravels, but its location only 21 m from the river and about 10 m above the river level suggests that it was also subject to abrasion during the Bonneville flood. Assuming a Bonneville surface age of 14.5 ka yields an accumulation rate of 13.5 μg cm$^{-2}$ ka$^{-1}$, at the lower end of the range of our previous results.

Normalized accumulation rates, $R_{NMn}$, are obtained by dividing the $N_{Mn}$ values by the corresponding exposure age (in ka) of the rock surfaces. They can be understood as a revarnishing rate, i.e., the rate (in percent per 1000 years) at which the varnish on a cleared surface approaches that of the surrounding intact varnish. In the case of the Melon Gravels, the present-day intact varnish represents 100%, and thus the revarnishing rate is 100% divided by 14.5 ka, equal to 6.9% ka$^{-1}$. Again, this value is somewhat lower, but still of comparable magnitude to those from our previous studies (10–14% ka$^{-1}$).

The internal consistency of these estimates can be tested by comparing the age estimates obtained using $R_{Mn}$ and $R_{NMn}$ for the petroglyphs at CP. A scatter plot of the ages based on $D_{Mn}$ ($R_{Mn}$) vs. those based on $N_{Mn}$ ($R_{NMn}$) is shown in Fig 2. A regression forced through zero (by definition, $D_{Mn}$ and $N_{Mn}$ on fresh surfaces are zero) yields a unity slope within statistical uncertainty ($r^2 = 0.71$), indicating that there is no significant bias between the two approaches.

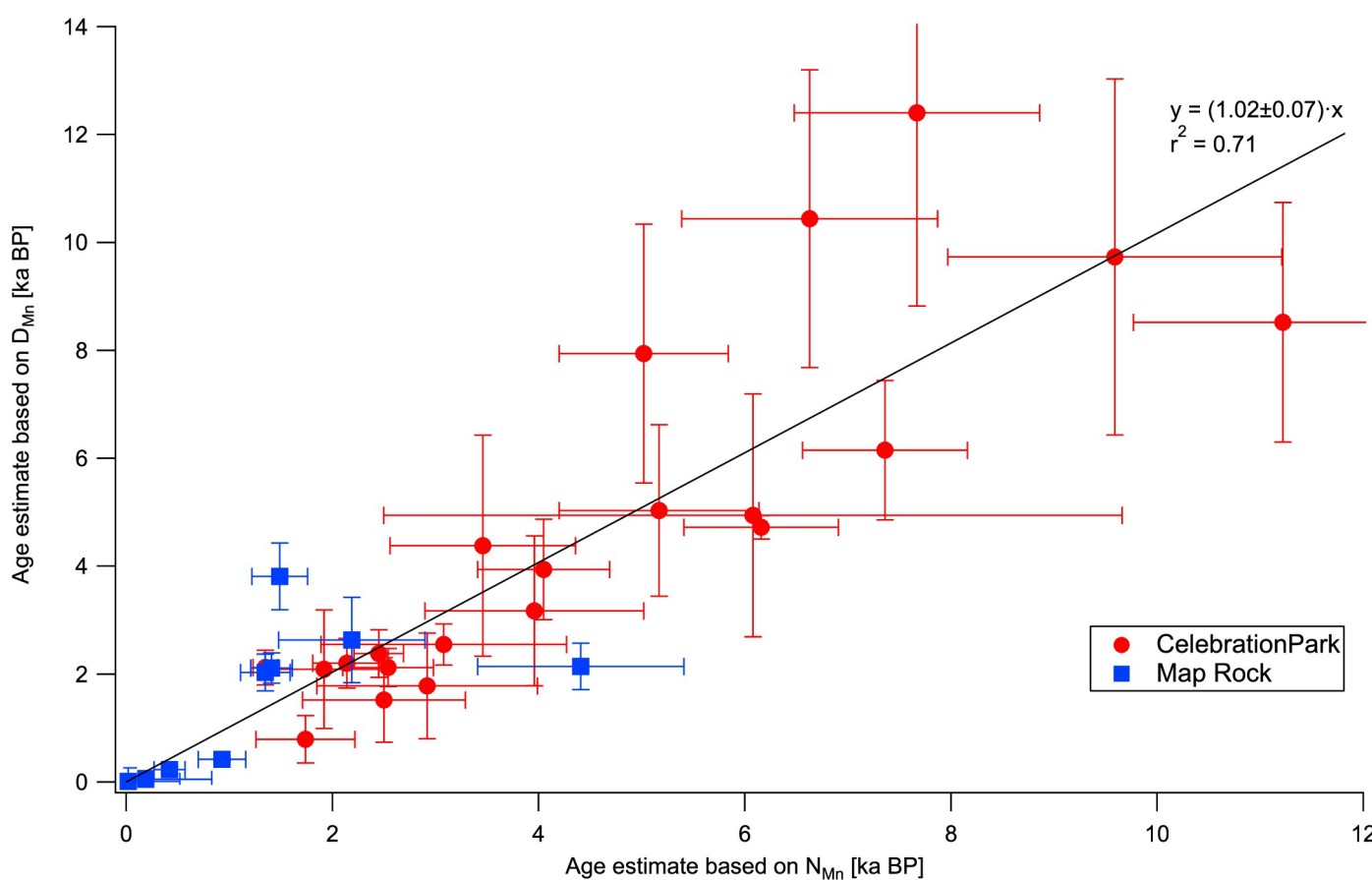

**Fig 2. Age estimates based on $D_{Mn}$ vs those based on $N_{Mn}$ for rock art elements at the Celebration Park and Map Rock sites.** The regression line is based on the data from Celebration Park. The error bars represent standard errors from the pXRF measurements and do not include the uncertainty of the age calibration.

## Rock varnish on petroglyph surfaces

### Celebration Park (10-CN-5)

A total of 86 panels have been recorded at this site by a team from the College of Western Idaho [51]. The rock art here is dominated by the Great Basin Pecked (GBP) style, with the largest number of elements (64%) in the Great Basin Rectilinear Abstract (GBRA) substyle, followed by Great Basin Curvilinear Abstract (GBCA, 29%) and Great Basin Representational (GBR, 7%). Dots or dot designs, circles, and circle designs are the most frequent motifs. The GBR motifs include anthropomorphs and reptilians, e.g., lizards and snakes, while bighorn sheep, a common motive elsewhere in the Great Basin [e.g., 99–101], are absent. Archaeological excavations at 10-CN-5 and adjacent sites yielded predominantly Late Archaic and a few Middle Archaic artefacts [52]. Images of exemplary elements are shown in Fig 3 and photographs of all studied petroglyphs with the age estimates based on our measurements are given in S1 File.

The results of the pXRF measurements on the rock art at CP are given in Table 3. The age estimates are plotted in Fig 4 in the order of increasing ages, with the exception of CP2198, which is a reworked part of CP 2201 and has been plotted together with that element. The results suggest that there has been continuous rock art creation at this site, starting in the Paleoindian Period and extending into the Late Prehistoric (the chronology used here follows that of D'Azevedo [16]). More than half of the ages fall into the Middle and Late Archaic, which is consistent with the archeological findings at CP. On the other hand, the rock art evidence suggests also significant human activity in the Early Archaic and Paleoindian Periods, for which no evidence was found in the excavations. This is, however, not unusual, as rock art sites often were sacred places, with little associated domestic activity.

The great antiquity of rock art in the northern Great Basin has been shown previously from sites in Oregon and Nevada, which yielded radiocarbon ages of ~12.5 to 8 ka BP [4]. A GB Curvilinear element from the Tom's Spring site in the Snake River Valley gave a VML age of 8.1 to 9.4 ka BP [102]. Consistent with the results from these other sites, our oldest motifs are in the Great Basin Abstract styles, and representational motifs only appear from the Middle Archaic onwards. This should, however, not be generalized to imply an evolutionary development from abstract to representational motifs, since at other sites, e.g., Legend Rock, representational motifs are among the most ancient rock art [see below, and 10].

Thus, our data suggest that the GB Pecked styles reached the northeastern GB by at least 10 ka ago. On the other hand, GBCA motifs, e.g., arch and dot designs, are also among the most recent elements, indicating that these symbols maintained their relevance to indigenous populations over a time span of some 10,000 years. Visual inspection of the petroglyphs shows evidence of reworking of some of the elements, in the form of heavily and lightly revarnished spots present in the same petroglyph. This is the case, for example, in the anthropomorph CP2201/CP2198 and the abstract pattern CP2146/CP2140. Even the reworked parts, however, show great ages and, in contrast to other sites, e.g., in the Mojave Desert [8] and the other sites in this study, there is no evidence for reworking in the Historic Period. In fact, all rock art activity at CP appears to have ceased around 1.5 ka BP. This may reflect a tendency to avoid rock art sites once they have been "re-socialized" through reworking by newly arrived populations [103].

### Map Rock (10-CN-10)

The most well-known petroglyphs at the Map Rock site are on a single isolated large boulder, but additional rock art is scattered on smaller boulders in its surroundings [53]. Map Rock was first described by Robert Limbert in a 1927 pamphlet created to promote tourism by railroad, who somewhat fancifully interpreted the petroglyphs as a detailed map of the Snake River

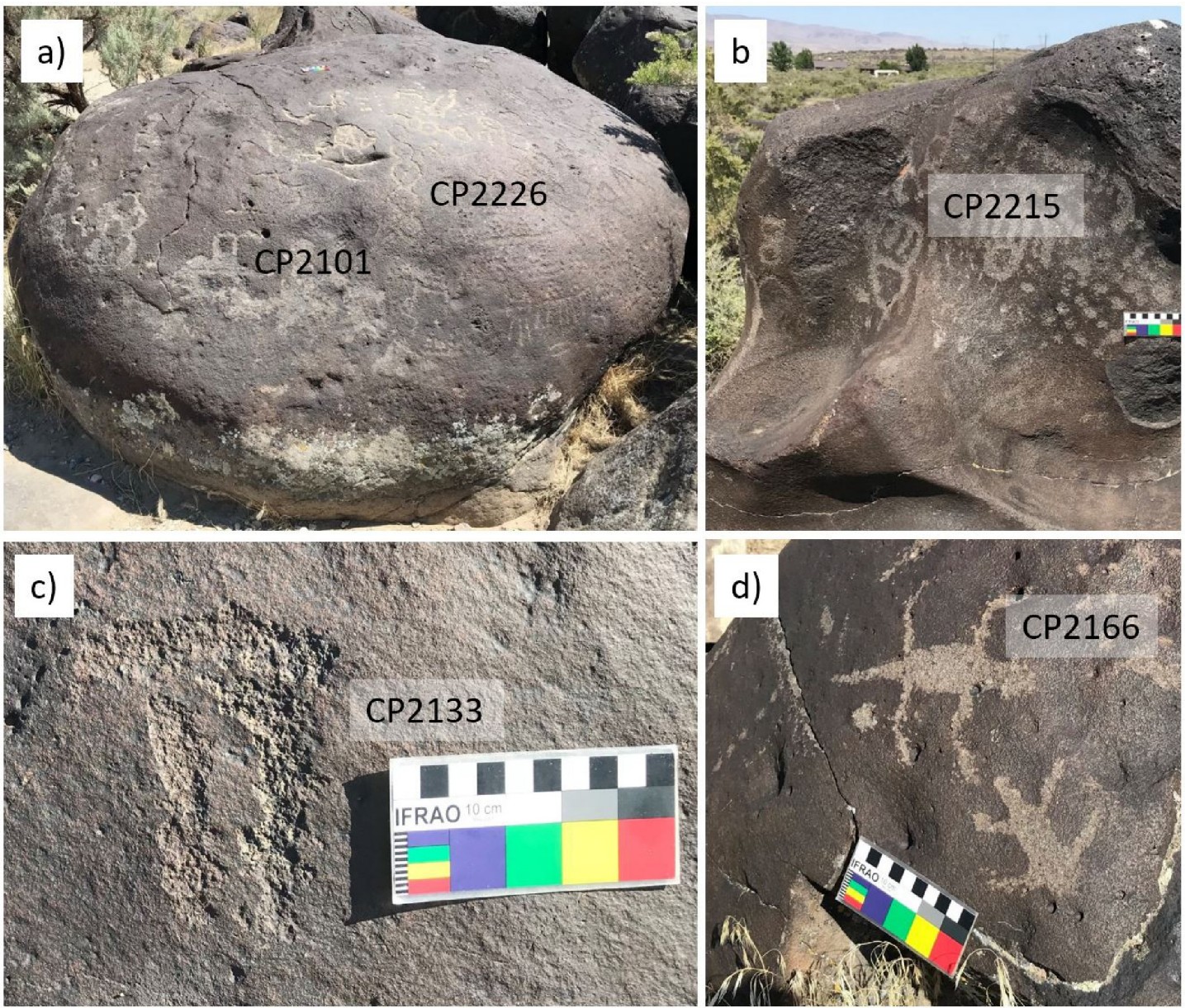

**Fig 3. Exemplary petroglyph images from Celebration Park.** (a) Melon Boulder with elements CP2101 and CP2226 (GBCA); (b) Complex abstract pattern (CP2215); (c) GB Rectilinear Abstract (CP2133); (d) Lizards, GB Representational (CP2166).

Valley [53]. The rock art at MR comprises a wide variety of motifs in the Great Basin Tradition, in particular the Abstract and Representational substyles. Representational motifs include anthropomorphs, hand prints, deer, bighorn and other quadrupeds, atlatls, and others; they account for 18% of the 147 elements mapped here. GBCA motifs (40%) are typified by wavy lines, tailed, spoked, sectioned and concentric circles, sun disks, and stars. GBRA motifs consist of meanders, rectangular grids, intersecting lines, rakes, dots and dot designs, chevrons, and zigzags [53]. A photograph of Map Rock with the locations of the visible investigated elements is shown in Fig 5 and photographs of all studied petroglyphs with the age estimates based on our measurements are given in S2 File.

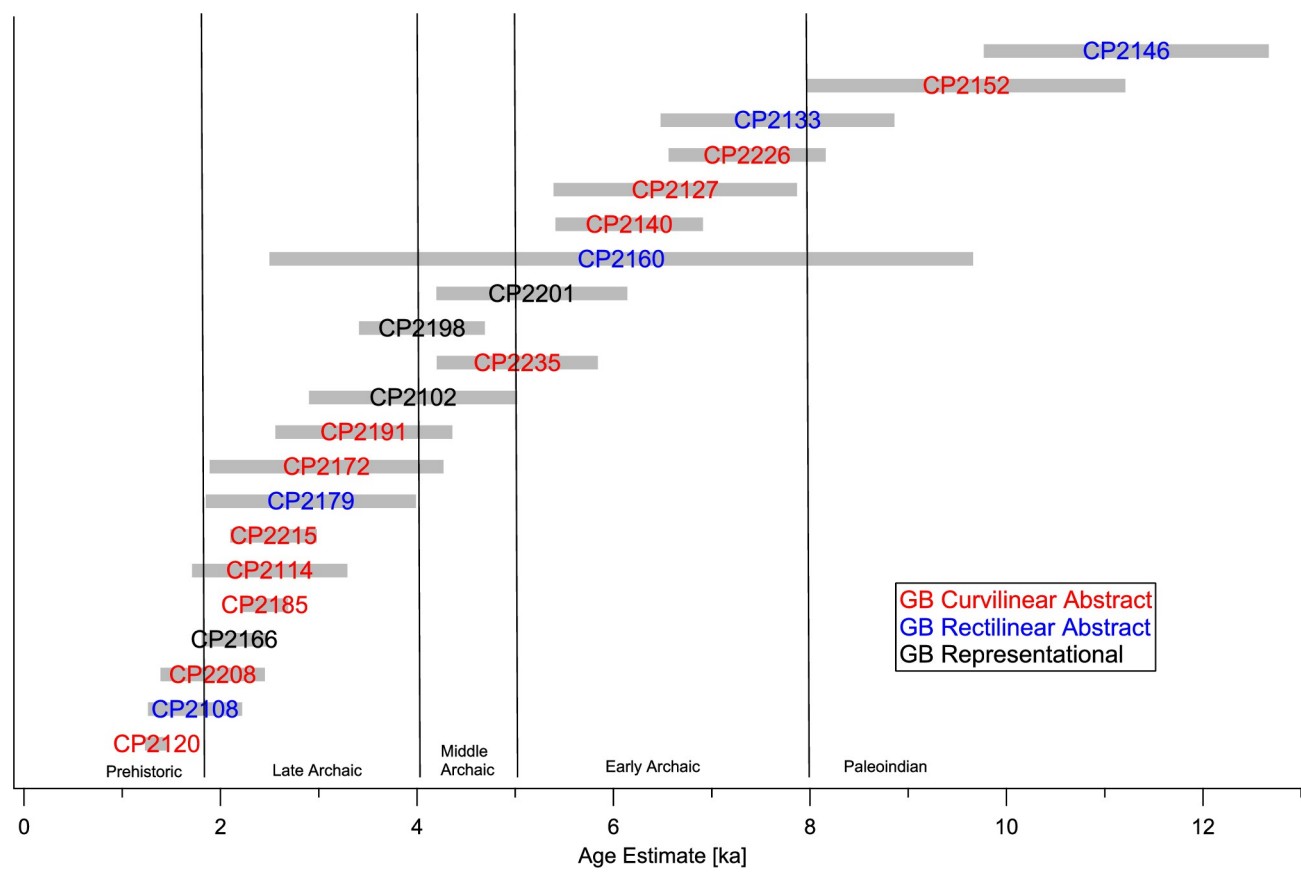

**Fig 4. Age estimates for the rock art elements at Celebration Park based on $N_{Mn}$ measurements.**

On MR, we conducted measurements on nine elements on the east, south, and west-southwest sides of the boulder. The results are shown in Table 4 and the age estimates are presented graphically in Fig 6. To derive age estimates at MR, we made the assumption that the boulder was fully cleared of any preexisting varnish during the 14.5 ka Bonneville flood event, and thus had an $R_{NMn}$ of 6.9% ka$^{-1}$, identical to CP. Given its location close to the Snake River, this seems a highly plausible assumption. In contrast to the rock art at CP, the elements at MR appear to be relatively recent, with most ages less than 2 ka. One exception is MR 2053, which gave an $N_{Mn}$ age of 4.4 ka. This value may be questionable, however, because this element is on the bottom of the main panel in the front of the boulder, which looks like it has been significantly abraded. Abrasion of the varnish surrounding the rock art in this area would give low $D_{Mn}$ values for the intact varnish, and consequently an erroneously high $N_{Mn}$ and age estimate for the rock art. This hypothesis is supported by the age calculated from the $D_{Mn}$ on this element, which is only 2.1±0.4 ka (shown as MR2053* in Fig 6), in good agreement with the age range estimated for the other elements at MR. Overall, the range of age estimates obtained for the rock art at MR is consistent with the archaeological excavations in this part of the Snake River Valley, which indicate human activity in the period from 2 ka BP to the present [52].

In contrast to CP, where all measurements showed pronounced varnish accumulation on all rock art elements, two elements at MR have no significant Mn accumulation. MR2078 is a deep indentation in the center of a complex abstract design on the front (south) side of the boulder. In a photograph from 1922, this design appears intact and no such indentation is seen (https://digital.boisestate.edu/digital/collection/Limbert/id/449/; last viewed 17 May 2021).

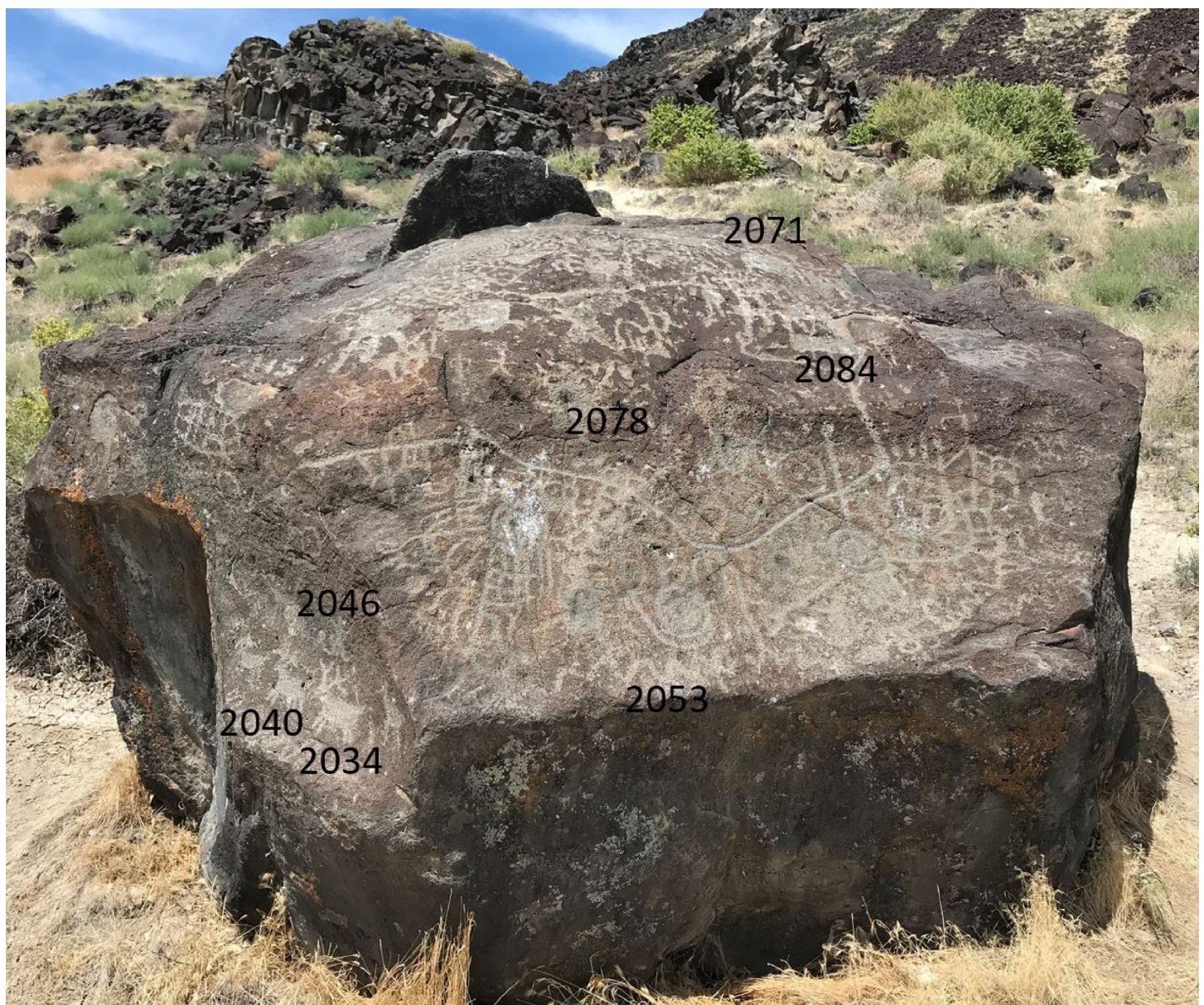

**Fig 5. South side of Map Rock with the locations of the visible investigated elements.**

Most likely, therefore, it is a bullet hole made some time in the last 100 years. The anthropomorph MR2046 showed a large degree of variability. It is closely associated with the two zoomorphs, MR2034 and MR2040, and its standard error range overlaps both of these elements, so that a synchronous production in the Prehistoric Period is most likely. Possibly, the apparent younger age of this element is the result of partial reworking.

## Legend Rock (48HO4)

In spite of its name suggesting a single rock, the Legend Rock site actually consists of a series of panels stretching some 500 m along a bluff. The rock art at LR has been discussed in considerable detail in several publications [11, 17, 71, 73, 74, 81, 104]. The most common styles

**Table 4. Mn absolute ($D_{Mn}$) and normalized areal densities ($N_{Mn}$) on the rock art elements at Map Rock and the corresponding age estimates based on $N_{Mn}$ [Avg.: Arithmetic average, S.D.: Standard deviation, S.E.: Standard error].**

| Element | Motif | Style | N | $D_{Mn}$ | | | $N_{Mn}$ | | Age estimate | |
|---|---|---|---|---|---|---|---|---|---|---|
| | | | | Avg. | S.D. | S.E. | Avg. | S.E. | Avg. | Error |
| | | | | [µg cm$^{-2}$] | | | [%] | | [ka BP] | |
| MR2078 | Abstract (bullet hole?) | GBCA | 3 | 0.1 | 5.8 | 3.4 | 0.1% | 3.4% | 0.0 | 0.5 |
| MR2046 | Anthropomorph | GBR | 3 | 0.6 | 3.7 | 2.1 | 1.3% | 4.4% | 0.2 | 0.6 |
| MR2034 | Elk | GBR | 3 | 3.1 | 1.6 | 0.9 | 2.9% | 1.1% | 0.4 | 0.2 |
| MR2040 | Bighorn sheep | GBR | 3 | 5.6 | 1.0 | 0.6 | 6.4% | 1.6% | 0.9 | 0.2 |
| MR2084 | Rectangle | GBRA | 3 | 27 | 7.9 | 4.6 | 9.3% | 1.6% | 1.4 | 0.2 |
| MR2066 | Bighorn sheep | GBR | 3 | 28 | 6.6 | 3.8 | 9.7% | 1.4% | 1.4 | 0.2 |
| MR2059 | Quadruped? | GBR? | 3 | 51 | 14 | 8.3 | 10% | 1.9% | 1.5 | 0.3 |
| MR2071 | Wavy line | GBCA | 4 | 35 | 21 | 10.6 | 15% | 4.9% | 2.2 | 0.7 |
| MR2053 | Wavy line | GBCA | 3 | 29 | 10 | 5.7 | 30% | 6.9% | 4.4 | 1.0 |

represented at LR are the Legend Rock Outline Complex (LROC), which consists largely of animal figures with a distinct outline, and the Dinwoody Anthropomorphs (DA), characterized by surreal anthropomorphic and therianthropic figures and associated animal motifs. In addition, there are some elements classified as En-toto Pecked (ETP) and some elements from the Historic Period, e.g., houses and names. Images of exemplary elements are shown in Fig 7

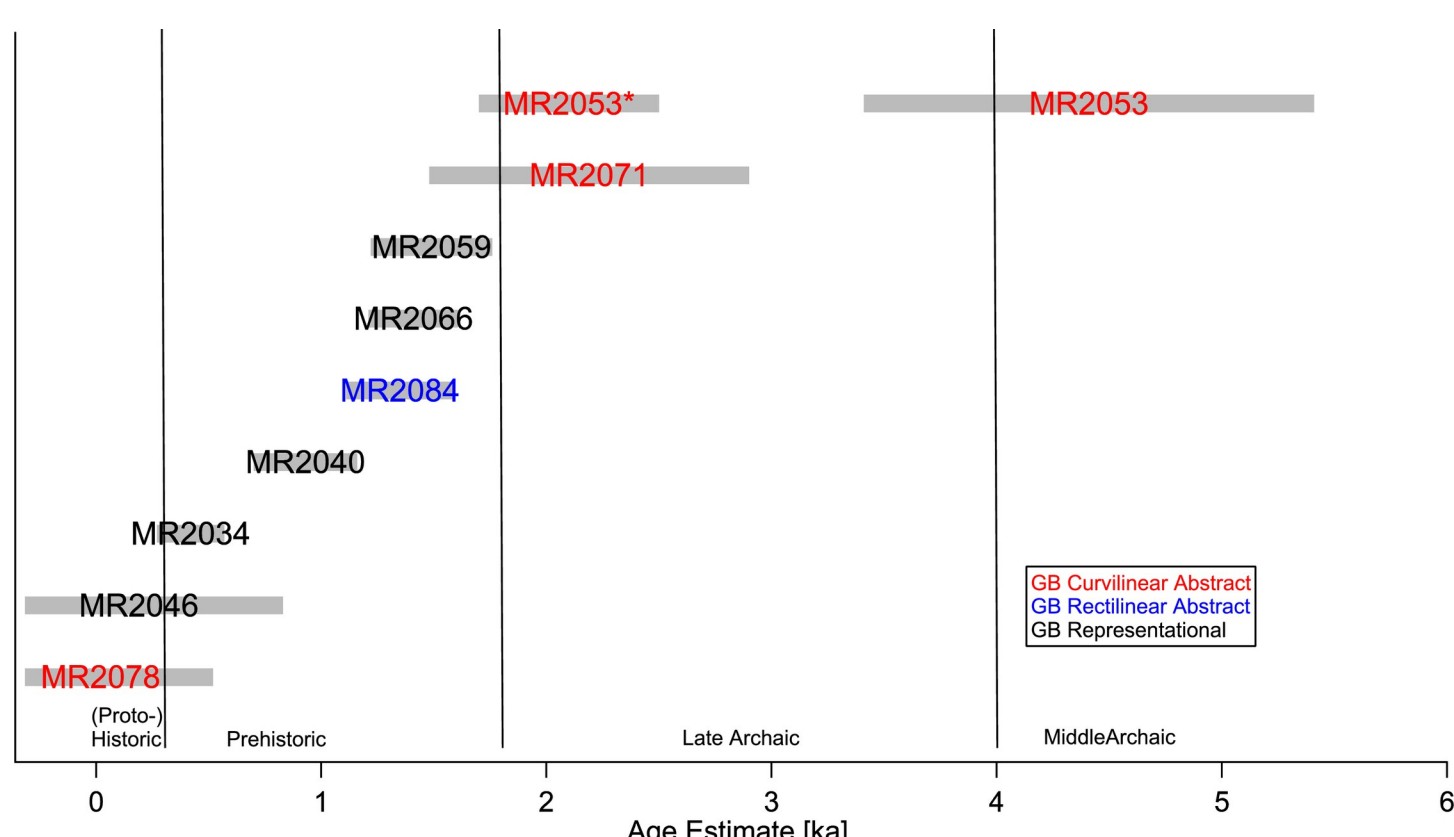

**Fig 6. Age estimates for the rock art elements at Map Rock based on $N_{Mn}$ measurements.** (MR2053* represents the age based on $D_{Mn}$ for this element).

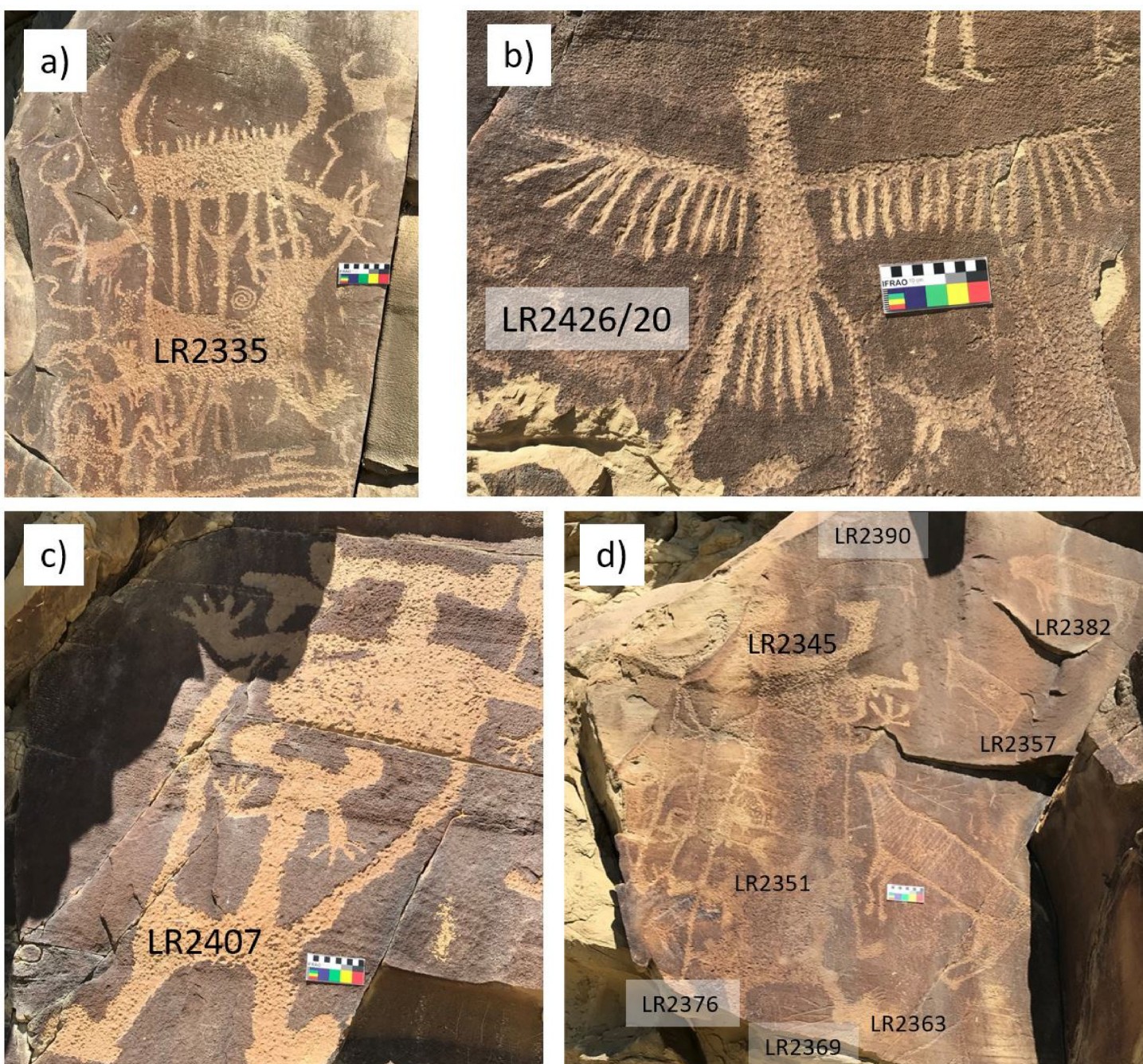

**Fig 7. Exemplary petroglyph images from Legend Rock.** (a) Dinwoody Anthropomorph; (b) Bird with some reworked areas; (c) Dinwoody Anthropomorph; (d) Panel 48 with Dinwoody Anthropomorphs and Legend Rock Outline Complex elements.

and photographs of all studied petroglyphs with the age estimates based on our measurements are given in S3 File.

We made measurements on 18 elements at LR, in two cases with multiple measurements on different parts of the same element. The results are listed in Table 5 and the age estimates

**Table 5. Mn absolute ($D_{Mn}$) and normalized areal densities ($N_{Mn}$) on the rock art elements at Legend Rock and the corresponding age estimates based on $N_{Mn}$ [Avg.: Arithmetic average, S.D.: Standard deviation, S.E.: Standard error].**

| Element | Motif | Style | N | $D_{Mn}$ | | | $N_{Mn}$ | | Age estimate | |
|---------|-------|-------|---|------|------|------|------|------|------|-------|
| | | | | Avg. | S.D. | S.E. | Avg. | S.E. | Avg. | Error |
| | | | | [μg cm$^{-2}$] | | | [%] | | [ka BP] | |
| LR2307 | Elk | LROC | 4 | 12.9 | 4.0 | 2.0 | 51% | 17.8% | 5.6* | 2.2 |
| LR2357 | Elk | LROC | 3 | 34 | 0.9 | 0.5 | 48% | 4.3% | 5.3 | 2.1 |
| LR2369 | Elk antlers | LROC | 3 | 75 | 9.6 | 5.6 | 78% | 10.4% | 8.6 | 3.4 |
| LR2382 | Bison | LROC | 4 | 39 | 4.2 | 2.1 | 63% | 5.5% | 7.0 | 2.8 |
| LR2390 | Artiodactyl | LROC | 3 | 47 | 3.1 | 1.8 | 45% | 3.0% | 5.0 | 2.0 |
| LR2393 | Bison | LROC | 3 | 57 | 8.9 | 5.2 | 55% | 5.8% | 6.1 | 2.4 |
| LR2400 | Bighorn | LROC | 4 | 75 | 6.1 | 3.0 | 62% | 4.2% | 6.8 | 2.7 |
| LR2335 | Dinwoody anthrop. | DA | 4 | 26 | 2.4 | 1.2 | 41% | 2.6% | 4.5 | 1.8 |
| LR2345 | Dinwoody anthrop. | DA | 3 | 53 | 7.6 | 4.4 | 40% | 3.8% | 4.4 | 1.8 |
| LR2351 | Dinwoody anthrop. | DA | 3 | 29 | 4.2 | 2.4 | 28% | 3.2% | 3.0 | 1.2 |
| LR2363 | Dinwoody anthrop. | DA | 3 | 30 | 9.0 | 5.2 | 33% | 8.2% | 3.7 | 1.5 |
| LR2376 | Dinwoody anthrop. | DA | 3 | 53 | 4.7 | 2.7 | 43% | 7.1% | 4.7 | 1.9 |
| LR2411 | Dinwoody anthrop. | DA | 3 | 23 | 2.1 | 1.2 | 20% | 3.9% | 2.2 | 0.9 |
| LR2407 | Dinwoody anthrop. | DA | 2 | 12.1 | 3.7 | 2.6 | 9.3% | 2.1% | 1.0 | 0.4 |
| LR2315 | Anthrop. or quadruped | DA | 3 | 1.1 | 0.4 | 0.2 | 6.3% | 2.5% | 0.7* | 0.3 |
| LR2437 | Rabbit | DA | 2 | 3.3 | 2.5 | 1.8 | 3.8% | 2.1% | 0.4 | 0.2 |
| LR2327 | Archer | ETP | 4 | 42 | 2.0 | 1.0 | 32% | 2.8% | 3.5 | 1.4 |
| LR2426 | "Thunderbird", original | ? | 4 | 29 | 4.3 | 2.1 | 39% | 7.4% | 4.3 | 1.7 |
| LR2430 | "Thunderbird", reworked | ? | 3 | 10.9 | 3.8 | 2.2 | 15% | 4.0% | 1.6 | 0.7 |
| LR2417 | House | HP | 2 | 1.0 | 1.4 | 1.0 | 2.6% | 2.6% | 0.3 | 0.3 |
| LR2419 | House | HP | 2 | 1.9 | 1.7 | 1.2 | 5.2% | 3.2% | 0.6 | 0.4 |

The age uncertainty estimate is based on an assumed uncertainty of 40% in $R_{NMn}$ or the measurement standard error, whichever is greater. (LROC: Legend Rock Outline Complex; DA: Dinwoody Anthropomorphs and associated zoomorphs; ETP: En-Toto Pecked; HP: Historic Period).

*) poorly developed varnish (low Mn/Fe) surrounding the element

graphically shown in Fig 8. In contrast to CP and MR, there is no calibration surface at LR with an a priori known age. In this situation, we made use of the fact that the $R_{NMn}$ in our previous studies had fallen into a relatively narrow range (10–14% ka$^{-1}$) and applied the average of our previous studies, i.e., 11% ka$^{-1}$, to our $N_{Mn}$ values from LR. This choice of $R_{NMn}$ is also supported by our measurements at Petroglyph Canyon (see below). As an uncertainty estimate for the age estimates, we used either an error of 40% to reflect the uncertainty in $R_{NMn}$ or the standard error of the $N_{Mn}$ values, whichever is greater. In the following, we discuss these age estimates in the context of estimates from previous work at LR based on other archaeometric and archaeological methods.

The LROC elements cluster consistently in the period around the Early Archaic, with the lower ends of the error bars reaching into the Middle Archaic and the upper ends reaching into the Paleoindian Period. This is in good agreement with the evidence from VML and other dating techniques, which indicates that this tradition may have begun around the TP/EH transition (12 to 11 ka BP) and, with some variations, may have lasted for thousands of years [9–11, 73]. Some of the elements in this group have been investigated by Francis and coworkers [74] using different techniques (VML, AMS, and CR). While all these techniques have their own issues, a comparison is nevertheless of interest. The elk antlers at the bottom of Panel 48 (LR2369, their WP90-10) gave an age estimate of 8.6±3.4 ka, a range that includes their CR age

**Table 6. Mn absolute ($D_{Mn}$) and normalized areal densities ($N_{Mn}$) on the rock art elements at Petroglyph Canyon and the corresponding age estimates based on $N_{Mn}$ [Avg.: Arithmetic average, S.D.: Standard deviation, S.E.: Standard error].**

| Element | Motif | Style | N | $D_{Mn}$ | | | $N_{Mn}$ | | Age estimate | |
|---|---|---|---|---|---|---|---|---|---|---|
| | | | | Avg. | S.D. | S.E. | Avg. | S.E. | Avg. | Error |
| | | | | [µg cm$^{-2}$] | | | [%] | | [ka BP] | |
| PC2480 | Quadruped | ETP | 4 | 21 | 8.5 | 4.3 | 31% | 8.1% | 2.8 | 1.1 |
| PC2493 | Anthrop. w/ feather | ETP | 3 | 10.4 | 3.4 | 2.0 | 23% | 5.4% | 2.1* | 0.8 |
| PC2485 | Anthrop. w/ headgear | ETP | 4 | 31 | 4.4 | 2.2 | 21% | 2.3% | 1.9 | 0.8 |
| PC2472 | Quadruped | ETP | 3 | 14.6 | 1.5 | 0.9 | 21% | 3.8% | 1.9 | 0.8 |
| PC2447 | Bison w/ blood | ETP | 3 | 23 | 4.8 | 2.8 | 16.4% | 2.9% | 1.5 | 0.6 |
| PC2455 | Blood | ETP | 3 | 10.0 | 0.3 | 0.2 | 7.3% | 0.9% | 0.7 | 0.3 |
| PC2501 | Archer | ETP | 3 | 3.4 | 0.4 | 0.2 | 9.8% | 6.4% | 0.9 | 0.6 |
| PC2458 | Thunderbird | ETP | 3 | 6.9 | 0.7 | 0.4 | 7.8% | 0.6% | 0.7 | 0.3 |
| PC2465 | Anthropomorph | ETP | 3 | 7.8 | 1.7 | 1.0 | 7.7% | 1.4% | 0.7 | 0.3 |
| PC2508 | Archer | ETP | 3 | 0.5 | 0.4 | 0.3 | 3.6% | 2.2% | 0.3* | 0.2 |

The age uncertainty estimate is based on an assumed uncertainty of 40% in $R_{NMn}$ or the measurement standard error, whichever is greater.

*) poorly developed varnish (low Mn/Fe) surrounding the element

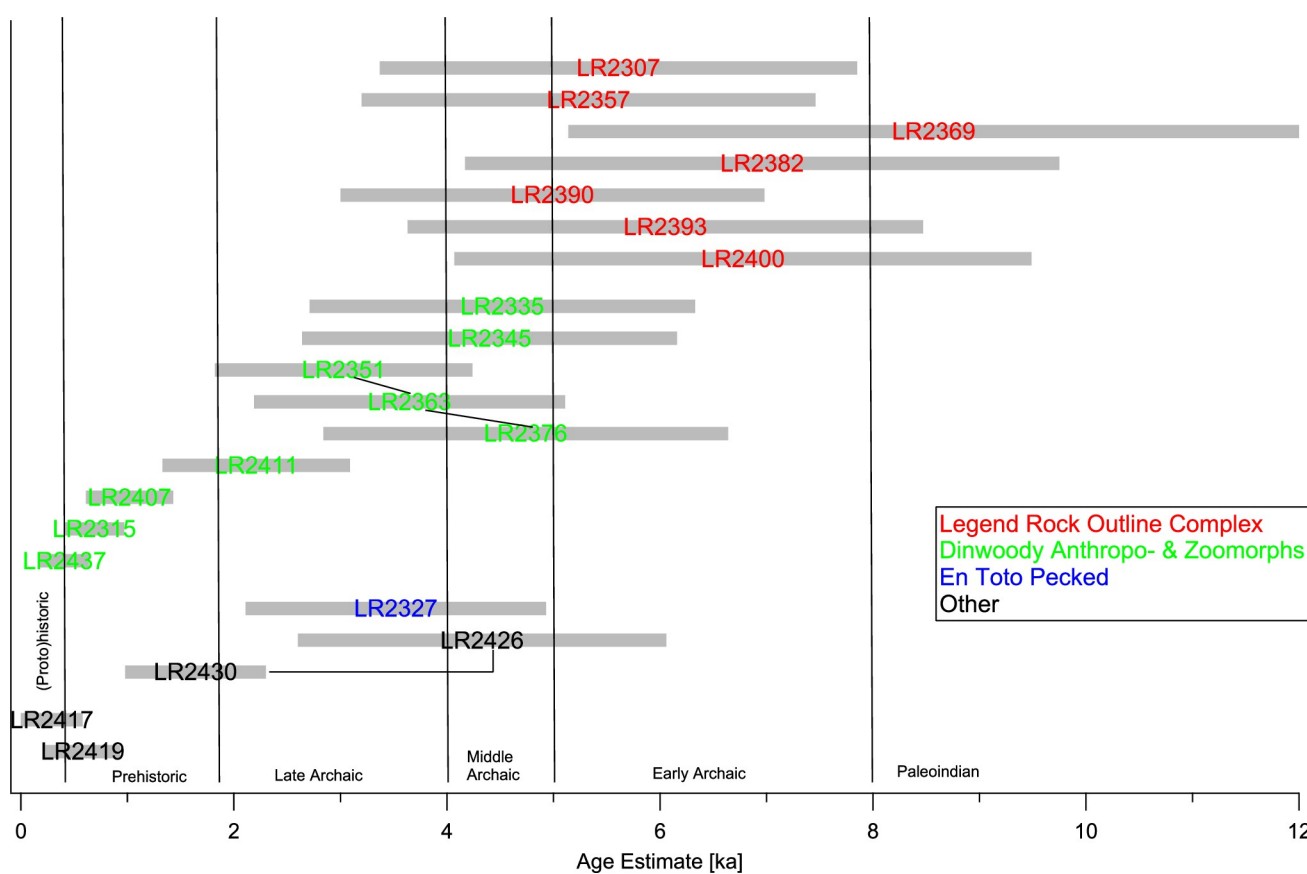

**Fig 8. Age estimates for the rock art elements at Legend Rock based on $N_{Mn}$ measurements.** Measurements on different parts of the same element are connected by lines. The age uncertainty estimate is based on an assumed uncertainty of 40% in $R_{NMn}$ or the measurement standard error, whichever is greater.

estimate of 6.2±0.8 ka [11]. The artiodactyl LR2390 (their WP90-4) gave 5.0±2.0 ka by our measurements, also overlapping their range of 6.5–7.0 ka from weathering rind AMS measurements. Finally, our estimate for the bighorn LR2400 [element F on Panel 35; 11] is 6.8±2.7 ka, in close agreement to their CR value of 6.8±1.8 ka. This panel, which is some 4 m above the present ground level, is probably one of the best preserved panels at LR and has not been subjected to the chalking and latex mold casting that has affected some of the other panels. VML data suggest that the oldest elements on this panel date to the LP/EH transition, around 11 ka BP. In summary, our measurements show an age range consistent with previous work by other techniques, suggesting that the LROC elements date in the period from the Late Paleoindian to the end of the Early Archaic.

The five investigated Dinwoody Anthropomorphs gave age estimates in the 1.0–4.7 ka range, in good agreement with the 1.0–5.4 ka range given for DAs and associated zoomorphs by Francis [11]. For two DA elements on Panel 48, we can make a direct comparison. Our LR2345 corresponds to WP90-5, with age estimates of 4.4±1.8 ka and 2.2±0.2, respectively. On WP90-6, which had a CR age of 2.7±0.3 ka, we made three measurements at different locations (LR2351/63/76), which gave 3.0±1.2, 3.7±1.5, and 4.7±1.9 ka. Overall, our pXRF age estimates appear to be somewhat higher than the earlier CR results, but in view of the uncertainty involved in both techniques, this difference cannot be considered significant.

The very unusual "rabbit" element, LR2437, showed an age of only 0.4±0.2 ka. Francis and coworkers [74] had obtained an AMS age of 295 a for this figure, but considered it too disturbed to be reliably dateable. This points to an additional complication with dating the rock art at LR. The varnish on the petroglyphs and the surrounding intact varnish on at least four of our studied elements may have been damaged by casting of latex molds and chalking. Latex molds were made in 1987 on the elk LR2357, the bison LR2382, and the rabbit LR2437, and chalking residues were noted in 2015 on the rabbit LR2437 and the thunderbird LR2436 [105]. We did not find obvious deviations in our age estimates for the latex-molded elements, possibly because the process did not remove enough varnish to affect our results. The "rabbit" has obviously a recent surface, but to what part this is the result of Indigenous or Euro-American reworking, latex-molding, or chalking cannot be resolved.

Similar considerations apply to the thunderbird LR2436 on Panel 74. Visual inspection clearly shows two types of surfaces: Most of the feathers and the higher points in the body are obviously abraded and less densely varnished, whereas the more deeply pecked areas, especially in the body and wings are much more strongly varnished. We made separate measurements on both types of surfaces and obtained age estimates of 4.3±1.7 ka for the original and 1.6±0.7 ka for the reworked surfaces. This would suggest that the image was originally created at about the same time as the Dinwoody figures on Panel 74 [11] and reworked by Indigenous people around the time that En-Toto Pecked rock art was created.

LR2315 is difficult to interpret–stylistically it may be a fully pecked Dinwoody zoomorph, but its content is unidentifiable. It yielded an age estimate of only 0.7±0.3 ka, but this value must be considered with caution as the surrounding coating had a very low Mn/Fe ratio and may not be true varnish.

One element in our data set appears to belong to the ETP style, the archer LR2327. The age estimate for this figure (3.5±1.4) is likely somewhat too high, as the bow and arrow was only introduced around 1.5–2 ka BP [66, 101, 106], highlighting the fact that age estimates on a single element using our approach must always be considered with caution.

Finally, there are two elements that are obviously from the Historic Period, the two houses LR2417 and LR2419. Their age estimates agree, within experimental error, with this period of production.

In summary, our measurements at LR suggest three major phases of rock art creation, which, however, show considerable overlap. The first, centered on the Early Archaic, is characterized by the large artiodactyls (elk, bison, bighorn) of the LROC complex. Second, the age estimates for the DAs and associated zoomorphs fall mostly into the Middle and Late Archaic, but extend into the Prehistoric Period. It is unclear, on the other hand, whether the youngest ages measured on the DAs represent the original creation of these elements in the Prehistoric to Historic Periods or reflect later reworking by Indigenous groups or Euro-Americans. Finally, the Prehistoric to Historic Periods saw the creation of ETP elements and images from the Euro-American culture, as well as reworking of some of the earlier rock art.

## Petroglyph Canyon (24CB601)

Petroglyph Canyon is the type locality for the ETP style, described in detail in Francis and Loendorf [17] and Loendorf [81]. The dominant motif consists of small, stylized, fully pecked anthropomorphs, often in groups. Zoomorphs are typically also highly stylized, with little detail, and without identifying species characteristics. Sites with ETP or similar elements are found across the Great Basin from Nevada to Wyoming. This style is likely associated with various Shoshone groups, who inhabited large parts of the Great Basin for 4000 years or more until they were displaced in the region of the ETP style by Crow Indians after about 1500 CE. There is a large peak in the frequency of radiocarbon dates between 2000 and 1000 BP, suggesting significant demographic and/or cultural changes around this time period [66].

While no EPT elements have been directly dated, the available evidence suggests that this style is relatively recent, ranging from about 2.6 to <1 ka BP, as indicated by radiocarbon and CR dating [11] and supported by radiocarbon dates from excavated cultural deposits at the site [81]. Further support for the anthropomorphs at the site dating from 1500 to 1000 BP comes from the depiction of bows and arrows, which replaced the atlatl around 2000–1500 BP [66, 101, 106]. Images of exemplary elements are shown in Fig 9 and photographs of all studied petroglyphs with the age estimates based on our measurements are given in S4 File.

Our pXRF data at PC come from nine elements, including a variety of human and animal figures (Table 6 and Fig 10). Similar to LR, there is no calibration surface at PC with an a priori known age. Consequently, we also applied the average of our previous studies, i.e., 11% ka$^{-1}$, to our $N_{Mn}$ values from PC, and used either an error of 40% to reflect the uncertainty in $R_{NMn}$ or the standard error of the $N_{Mn}$ values, whichever is greater, as uncertainty estimate. This allows us to compare our pXRF age estimates with previous estimates obtained using either weathering-rind AMS or CR dating [11, 74].

Of particular interest is the image of a bison, from whose mouth blood appears to emanate PC2447 = WP-90-32). Using our $R_{NMn}$ value of 11% ka$^{-1}$, we obtain an age estimate of 1.4±0.6 ka, in excellent agreement with the AMS age of 1470±75 [74]. If we had inverted our approach and used the AMS age of Francis et al. as calibration point to deduce $R_{NMn}$, we would have obtained a value of 11.6% ka$^{-1}$, which provides some internal consistency support to our choice of $R_{NMn}$. The blood coming out of the bison's mouth gave an age estimate of 0.7±0.3, also consistent with their CR estimate of <1.0 ka. Our measurements on two quadrupeds gave estimates of 2.8±1.1 and 1.9±0.8 ka, which compare favorably with CR ages of 2.6±0.4 and 2.5±0.3 ka on two (different) quadrupeds at PC. Our "thunderbird" PC2458 yielded an age estimate of 0.7±0.3 ka, also consistent with the CR result of an age <1.0 ka. Other elements compare less well: Our anthropomorphs PC2493 and PC2485 suggest significantly higher ages (2.1±0.8 and 1.9±0.8 ka) than the corresponding AMS and CR estimates (1.25±0.07 and 1.0±0.2 ka, respectively). We caution, however, that our estimate for PC2493 is of low confidence, since the varnish on this surface has a very low Mn/Fe ratio and low $D_{Mn}$ of both the petroglyph and

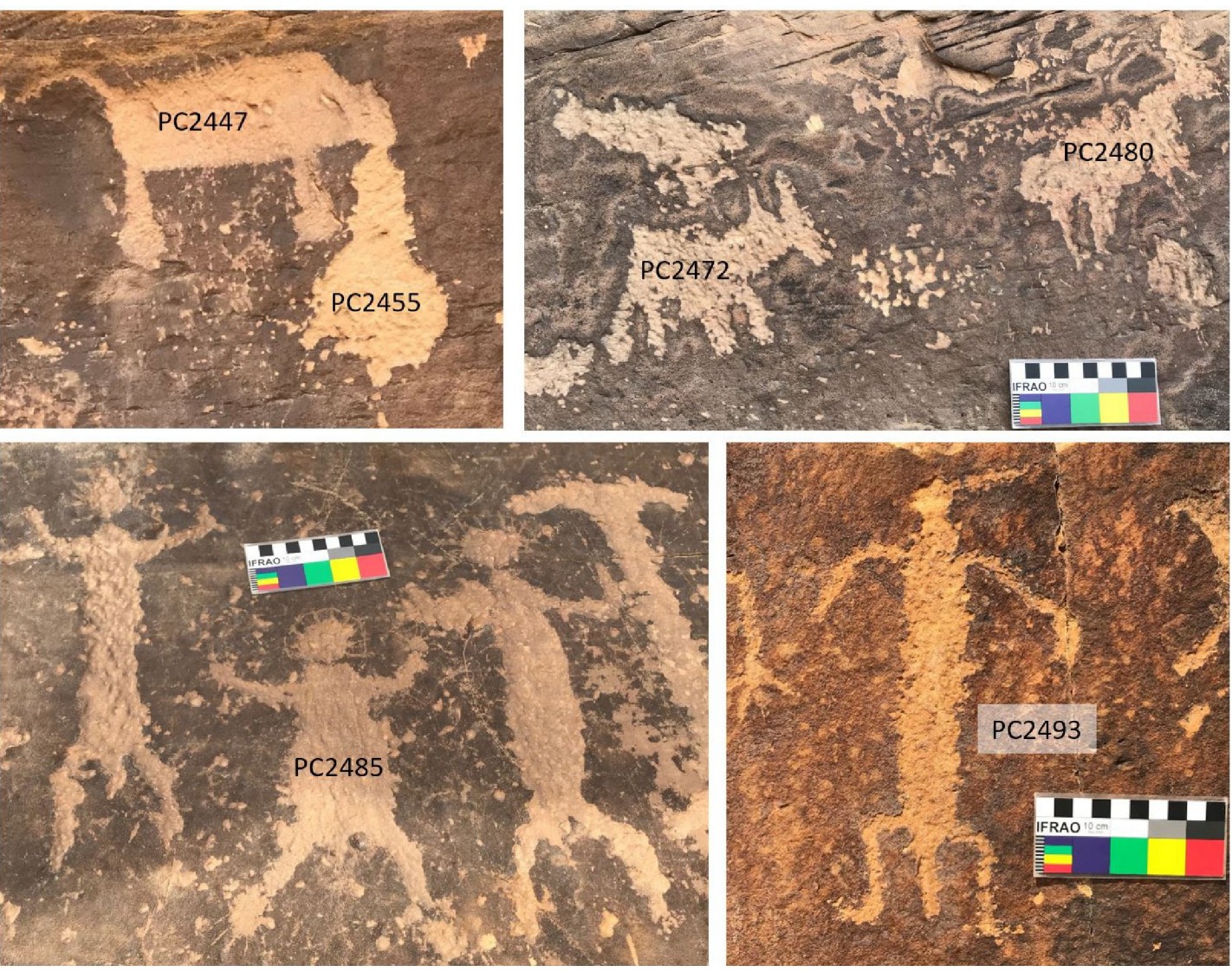

**Fig 9. Exemplary petroglyph images from Petroglyph Canyon.** (a) En-Toto Pecked bison with blood coming from its mouth; (b) ETP quadrupeds; (c) Group of ETP anthropomorphs; (d) ETP anthropomorph with feather headgear.

surrounding varnish, indicating that the rock surface is not well suited to varnish development.

Our estimate for the anthropomorph PC2465 (0.7±0.3 ka) is lower than the CR estimate of 1.3±0.1 ka for the same figure (WP90-31). The archers PC2501 and PC2508 fall into the time period when this weapon had come into use, but similar to the situation with PC2493, we caution that the varnish at PC2508 has unusually low Mn content.

In summary, we find that the age estimates obtained at LR are in reasonable agreement with those previously obtained with other methods and with the archeological evidence for the time of human presence at the site. Differences seen for individual elements do not exceed what would have to be expected given the methods' uncertainties, and would likely not be considered significant if more conservative error estimates had been assigned to the CR and AMS age estimates. Rock art creation at PC took place predominantly in the Prehistoric Period and

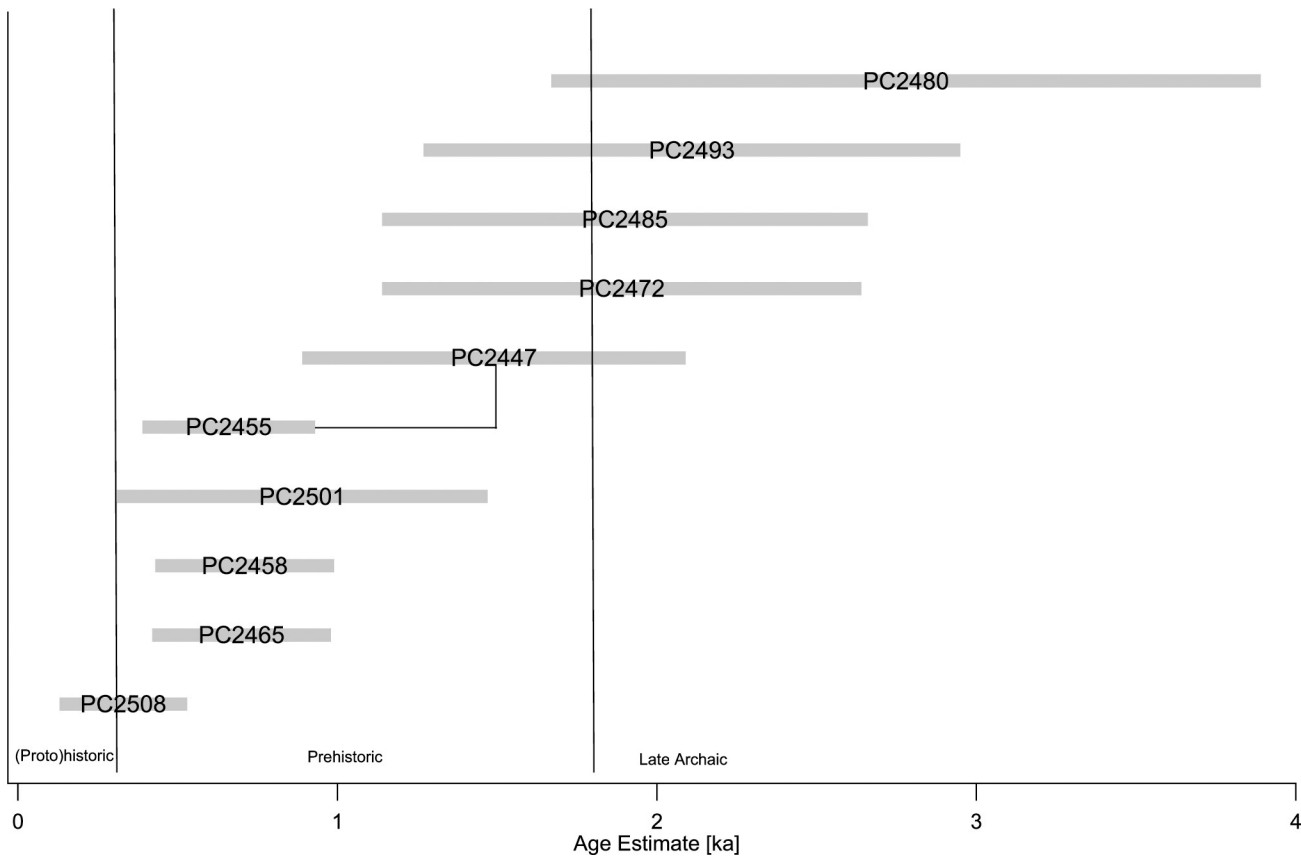

**Fig 10. Age estimates for the rock art elements at Petroglyph Canyon based on $N_{Mn}$ measurements.** Measurements on different parts of the same element are connected by lines. The age uncertainty estimate is based on an assumed uncertainty of 40% in $R_{NMn}$ or the measurement standard error, whichever is greater.

possibly towards the end of the Late Archaic, with no evidence for significant activity in the post-contact period. The single exception is a drill hole for uranium prospecting, where we found a $D_{Mn}$ below the detection limit.

## Summary and conclusion

We determined the areal density of Mn ($D_{Mn}$) and Fe ($D_{Fe}$) in rock varnish on dated Late Pleistocene basalt boulders and Holocene lava flows, on petroglyphs at four sites, and on intact rock varnish adjacent to the petroglyphs, using portable in-situ X-ray fluorescence spectrometry. Our sites were located in the Snake River Valley of Idaho and the Wind River Basin of western Wyoming and adjacent southern Montana.

The $D_{Mn}$ and $D_{Fe}$ values at these sites were lower than at our previous sites in the Mojave Desert and in Saudi Arabia. The average $D_{Mn}$ of intact varnish on the basalts in the Snake River Valley was 390±210 µg cm$^{-2}$, compared to our value of 630±320 µg cm$^{-2}$ on the Mojave Desert basalts. This may be in part due to the younger age of the Snake River basalts, but also to a wetter climate at these sites compared to the Mojave Desert. On the sandstone surfaces in the Wind River Basin, we measured an average $D_{Mn}$ of 79±44 µg cm$^{-2}$, also distinctly lower than the site averages from our five Arabian sites, which ranged from 105 to 209 µg cm$^{-2}$. Differences in climatic conditions and in the weathering resistance of the sandstones are likely factors that could account for the lower varnish densities at our Great Basin sites. The average

standard error for the $D_{Mn}$ determinations from replicate measurements on individual petroglyphs and surrounding intact varnish was 14 and 13%, respectively.

The ratio $D_{Mn}/D_{Fe}$ can be interpreted as the mean Mn/Fe mass ratio in the rock varnish. We use this ratio as an indicator of the presence of true rock varnish as opposed to other surface coatings and as a rough proxy for the formation environment of the varnish. The average Mn/Fe ratios at our sites ranged from 0.25 to 0.81, within the range typical of true rock varnish (0.2–1.3). They are in the lower part of this range, suggesting a formation under relatively dry conditions. However, some of the measurements, especially at PC, showed unusually high $D_{Fe}$ values and consequently had Mn/Fe ratios well below the range typical of true rock varnish. Visually, these coatings were reddish, in contrast to the dark brown to black color of true rock varnish. Such Fe-rich coatings are formed in environments where the physicochemical conditions are not conducive to Mn-oxyhydroxide deposition, e.g., extremely dry locations or at the underside of boulders, and are thus not likely to yield reliable age estimates with our technique.

The lifetime-averaged Mn accumulation rate in the form of varnish on a rock surface can be calculated from the measured $D_{Mn}$ if the exposure age of the rock surface is known. At our sites, there were two types of surfaces available for which this condition is satisfied, the "Melon Gravel" boulders in the Snake River Valley, which had been extensively abraded during the Bonneville Flood event 14.5 ka BP, and Holocene lava flows in Craters of the Moon NM, which had formed around 2.1 to 2.4 ka BP. From these surfaces, we obtained $R_{Mn}$ values of 28.2±9.9 and 25.6±12.4 $\mu g\ cm^{-2}\ ka^{-1}$, respectively, comparable to values we had found previously on basalts in the Mojave Desert. The good agreement between the two values suggests that our underlying assumption of a nearly linear growth rate over the Holocene for the varnish on basalt is not unreasonable.

To derive age estimates for rock art elements, one can simply divide the measured $D_{Mn}$ on the petroglyph by the Mn accumulation rate. However, because the accumulation rate on a specific surface is also affected by factors other than exposure age, we prefer to use the normalized Mn density, $N_{Mn}$, and the corresponding normalized accumulation rate, $R_{NMn}$, which reduces the effect of these confounding variables. On the Melon Gravels, we determined an $R_{NMn}$ of 6.9% $ka^{-1}$, somewhat lower than our previous results. For the sandstone surfaces, we made use of the fact that the $R_{NMn}$ on sandstone in our previous studies had fallen into a relatively narrow range (10–14% $ka^{-1}$) and used the average of these studies, i.e., 11% $ka^{-1}$, to derive age estimates.

Because we have no absolutely dated sandstone surfaces at our sites, we were not able to calculate absolute Mn accumulation rates on this substrate. However, a rough estimate for the Mn accumulation rate on the sandstone surfaces can be obtained by making the assumption that the $D_{Mn}$ based ages should be equal to those based on $N_{Mn}$. The $R_{Mn}$ that satisfies this constraint is then the product of $R_{NMn}$ (11 $ka^{-1}$) and the regression slope of $D_{Mn}$ vs. $N_{Mn}$ from the petroglyph measurements. Pooling the data from LR and PC yields a value of 1.06±0.08 $\mu g\ cm^{-2}$ (N = 30, $r^2$ = 0.83) for this slope, resulting in an estimate of 11.7±0.8 $\mu g\ cm^{-2}\ ka^{-1}$ for $R_{Mn}$, slightly below our Arabian values of 13.4–17 $\mu g\ cm^{-2}\ ka^{-1}$.

Based on our $N_{Mn}$ and $R_{NMn}$ data, we derived age estimates for the rock art elements at our four petroglyph sites. We emphasize that these dates should be considered experimental, and that because of the effects of non-age-related influences on Mn accumulation, the estimates on individual elements are subject to significant uncertainty, as expressed by the error bars in our plots. However, as discussed in the section on variability, these confounding effects tend to cancel with increasing number of observations, so that the age estimates on related groups of elements and motifs are much more robust.

The greatest ages were obtained at Celebration Park, where our estimates ranged from 11 to 1.4 ka BP, i.e., from the Paleoindian to the Prehistoric Periods. The dominant styles at this site, Great Basin Curvilinear and Rectilinear Abstract, represent the earliest images, but these styles continue to be used into the Prehistoric Period. Great Basin Representational images only appear from the Middle Archaic onwards, but do not replace the Abstract styles. This finding should not be construed as evidence for an evolutionary development from abstract to representational motifs, however, since at other sites representational motifs are among the most ancient rock art. The age range for the rock art at this site agrees with estimates for petroglyphs at other sites in the region, and is also consistent with the time over which indigenous populations have been present in the northern Great Basin, i.e., since about 16 ka BP. This suggests that rock art was part of the culture of the earliest inhabitants of North America, who might have even brought it with them from northeastern Asia.

The Map Rock site presents a complex set of petroglyphs in the GB Abstract and Representational Styles, with age estimates from the Late Archaic (ca. 2 ka BP) to the Historic Periods. In contrast to CP, where there was no evidence for recent damage to the petroglyphs we investigated, at MR there was evidence of recent vandalism in the form of a bullet hole and total lack of varnish on at least one of the elements.

Like at CP, the rock art at Legend Rock displays great time depth, potentially reaching back to the transition from the Paleoindian to the Archaic Periods (ca. 8 ka BP), and continuing to the Protohistoric Period. Here, there appears to be distinct progression of styles, with the Legend Rock Outline Complex (mostly zoomorphs) in the Early and Middle Archaic Periods, and the Dinwoody Anthropomorphs and associated zoomorphs predominantly in the Middle and Late Archaic, with some potential reworking into the Protohistoric. Indigenous reworking, but also chalking and latex casting in the 20[th] century, may have resulted in the recent apparent ages found on some of the elements. We found good agreement between our age estimates and previous estimates from other techniques at this site.

Petroglyph Canyon is the type locality for the En-Toto Pecked Style, which dates from the Late Archaic to the Protohistoric periods. The age estimates obtained here are in reasonable agreement with previous ones made with different techniques and are also consistent with some of the image content, e.g., humans armed with bows and arrows. Like at the other sites, there is evidence for more recent reworking or additions, e.g., the blood that appears to emanate from the mouth of a killed bison.

Looking at all the sites studied here, we find great age depth and continuity for the rock art creation in this part of the Great Basin, in the case of CP reaching across some 11,000 years from the Pleistocene/Holocene transition to the recent past. Subsequent human populations added to and modified rock art at these sites over thousands of years. Reworking of particular images is evident at all sites, and the presence of significant varnish on most reworked surfaces is clear evidence that this was done by Indigenous people before the arrival of Euro-Americans.

Like in our previous work from North America and Arabia, we find that the age estimates we obtained with our technique are consistent with the archeological and iconographic evidence as well as with the dates obtained with other techniques at these sites. Substantial uncertainties in the individual age estimates remain, however, caused by both the influence of a variety of local characteristics, e.g., exposure and weathering resistance of the rock surface, and the paucity of calibration surfaces with known ages. The few cases with very large errors might be identified by obvious age disagreements between iconographically similar images, by age estimates outside of the archaeologically possible range, or from strong disagreement between ages based on absolute and normalized Mn densities (e.g., MR2053, Fig 6). The effect of smaller errors, on the other hand, can be mitigated by increasing the number of petroglyphs

analyzed and focusing more on the ages obtained for groups of iconographically related elements than for individual images.

Future studies might benefit from including a wider range of dated lava flow surfaces, but accumulation rates from these surfaces may not be applicable to sandstone. In spite of these limitations, however, we feel that our approach has the unique potential to obtain age estimates for a large number of rock art elements with a reasonable amount of effort and, importantly, without the need for destructive sampling.

## Supporting information

**S1 Fig. Overview map of the study area.** (Map services and data available from U.S. Geological Survey, National Geospatial Program).
(PDF)

**S1 File. Images of the petroglyphs measured in Celebration Park.**
(PDF)

**S2 File. Images of the petroglyphs measured at Map Rock.**
(PDF)

**S3 File. Images of the petroglyphs measured at Legend Rock.**
(PDF)

**S4 File. Images of the petroglyphs measured in Petroglyph Canyon.**
(PDF)

## Acknowledgments

We acknowledge that this research was conducted on the ancestral lands of the Shoshone Indigenous People. We thank the Canyon County Parks Programs Manager, Juli McCoy, for permission to conduct the measurements at Celebration Park and Map Rock and the Wyoming State Parks Superintendent, Kevin Skates, for permission for the work at Legend Rock. We thank Julie Francis and Larry Loendorf for valuable discussions, and Larry Loendorf for taking us to Petroglyph Canyon. This research was supported by the German Max Planck Society and King Saud University, Riyadh, Saudi Arabia.

## Author Contributions

**Conceptualization:** Meinrat O. Andreae.

**Data curation:** Meinrat O. Andreae.

**Investigation:** Meinrat O. Andreae, Tracey W. Andreae.

**Methodology:** Meinrat O. Andreae.

**Writing – original draft:** Meinrat O. Andreae.

**Writing – review & editing:** Meinrat O. Andreae.

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
