## [Decision Letter · Decision Letter 0]

2 Nov 2021

PONE-D-21-23506Archaeometric studies on rock art at four sites in the northeastern Great Basin of North AmericaPLOS ONE

Dear Dr. Andreae,

Thank you for submitting your manuscript to PLOS ONE. After careful consideration, we feel that it has merit but does not fully meet PLOS ONE’s publication criteria as it currently stands. Therefore, we invite you to submit a revised version of the manuscript that addresses the points raised during the review process.

Dear Authors

The Manuscript Number PONE-D-21-23506 titled "Archaeometric studies on rock art at four sites in the northeastern Great Basin of North America" represents a very ambitious paper and I guess in can bring important advances to the chronological studies of rock artifacts. Unfortunately I had very contrasting reviews from 3 different scientists working in the field. The most negative one raised up some comments onto your work. I guess that working in the field of archaeometry should be intended also as trying to accomplish the (very difficult) task to syntesyze correctly applicative and formal (in scientific sensu) aspect of performed studies. So I would like to ask you to rehandle the paper in order to assess the issues highlighted mostly by reviewer 2 and at a minor entity by reviewer 3.

We look forward to receiving your revised manuscript.

Kind regards,

Fabio Marzaioli, Ph.D

Academic Editor

PLOS ONE

 “The author(s) received no specific funding for this work”

Reviewers' comments:

Reviewer's Responses to Questions

**Comments to the Author**

1. Is the manuscript technically sound, and do the data support the conclusions?

Reviewer #1: Yes

Reviewer #2: No

Reviewer #3: Yes

2. Has the statistical analysis been performed appropriately and rigorously? 

Reviewer #1: Yes

Reviewer #2: No

Reviewer #3: Yes

3. Have the authors made all data underlying the findings in their manuscript fully available?

Reviewer #1: Yes

Reviewer #2: Yes

Reviewer #3: Yes

4. Is the manuscript presented in an intelligible fashion and written in standard English?

Reviewer #1: Yes

Reviewer #2: Yes

Reviewer #3: Yes

5. Review Comments to the Author

Reviewer #1: Nice work. In particular the authors fully described the pitfalls of the technique. Might help to give a better idea about "significant uncertainty" in absolute dating. I see two types of error - things gone completely wrong and things working, subject to the many expected small perturbations in conditions.

I have a bit of concern about their use of the calibration curve from the cathedral study [83]. Were their varnish thicknesses and Mn content similar to those in that study?

Also, did the non-linear attenuation for the rock Mn-content come into play with the straight rock background correction?

Reviewer #2: This paper proposes a method to date ancient petroglyphs based on measurements by p-XRF of the areal densities on manganese and iron. The idea is based on the concept that the amount of varnish that has regrown on a petroglyph since its creation, relative to the surrounding intact varnish, is a measure of its age.

It seems a promising manuscript from the title and abstract, but the content lacks rigour since there are statements that evidence authors’ poor knowledge of XRF analysis. The method has been already published ([46, 47]). Moreover, the experimentation is very poor: using a single spectroscopic technique is insufficient to characterize a material. The authors they-self state : “We emphasize that these approaches remain experimental and have to be viewed with great caution, because the growth rate of varnish is highly variable and depends on a large number of parameters other than age, including the exposure of the rock surface to dust, erosion by wind and water, the orientation and slope of the rock surface, the hardness, roughness, and texture of the rock underneath, and its initial iron content, as summarized in Andreae, Al-Amri [7, and references therein]”. Because all of this, I recommend rejecting this paper.

My specific concerns are the following:

1) Introduction Section: It is hard to find out how this paper is advancing the state-of-the-art since it is not explicitly clarified in the paper. I suggest the authors dedicate a section to explain this.

2) Material and Methods Section: why the four archaeological sites have been chosen is not provided? The sentences “The pXRF measurements were conducted using a Niton XL3 pXRF (Thermo Fisher Scientific) in the “mining” mode” and “The pXRF measurements, reported as percentage by the instrument” are unacceptable in a scientific paper.

3) Results Section: the main issue is the confidence in the areal densities. They are deduced using experimental and quantitative methods that are openable and not so challenging. This argument lacks the required knowledge in XRF analyses.

Finally, I have observed that most of the methods used are not developed by the authors but taken from the literature. Specifically, this method is not new (others applied it before [46, 47]). Moreover, the p-XRF method is not improved in this paper but applied to solve their problem. So there is no scientific contribution in this aspect either.

As for the archaeological, I’m not able to judge if the results are worth to be published. Surely, whatever the historical and artistic repercussions, the authors must be sure that the experimental data on which their considerations are based are valid. This is not the case.

Reviewer #3: In this paper, the possibility of deriving age estimates from measurements of the areal density of manganese (DMn) and iron (DFe) in rock paint on petroglyphs was explored. It is based on the fact that the amount of paint that has grown back on a petroglyph since its creation, relative to the surrounding intact paint, is a measure of its age. To determine the areal densities of Mn (DMn) and Fe (DFe) in the rock paint, XRF measurements were conducted with a hand-held spectrometer on petroglyphs located in the Snake River Valley of southern Idaho and the Wind River Basin of western Wyoming and adjacent to southern Montana. In addition, XRF measurements were made on the intact paint surrounding rock art to assess the variability of paint formation between different rock surfaces and on standards (rock surfaces of known geological age) to assess their potential to derive accumulation rates of paint. In conclusion, the age estimates of the petroglyphs were derived and evaluated in their archaeological and cultural context. The results, in most cases, are in good agreement with chronologies based on archaeological and other archaeometric techniques.

The authors are clearly aware of the limitations of the method and the results, but explain the reasons effectively. This article could be a starting point for non-destructively dating important historical artifacts.

The experiments, statistics and other analyzes performed are described in sufficient detail. The sample size (461 measurements) is adequate to produce robust results. While the method is subject to significant uncertainty regarding the absolute age of individual images.

The conclusions are presented appropriately and are supported by the data. Where this was not possible, a reasonable discussion was reported.

The article is presented in an understandable way and is written in standard English.

The authors are not new to this field; in previous work on the paint of rocks in Arabia and North America, they showed that the age estimates obtained (determining the areal density of Mn (DMn) and Fe (DFe)) were consistent with the ages based on the cultural and ecological content of the art rock and allowed a significant ordering of the rock images in a sequence of ages. Therefore, the presented study replicates and is very similar to the previous work, but the authors provided solid scientific rationale for the presented work by clearly referencing and discussing existing literature.

In conclusion, in my opinion the document is suitable for publication and requires only a minor revision.

Specific comments:

The materials part should be reduced.

The pXRF instrumentation part should be better described. For example, what it means "Filter passes and integration periods were:" standard "25 s," low "15 s," high "20 s, and" light "25 s." (lines 300 and 301) What's the difference?

Line 385 The value on this flow surface is 0.81 ± 32, is the error correct?

Line 809 “rack varnish” correct it into “rock varnish”.

6. PLOS authors have the option to publish the peer review history of their article (what does this mean?). If published, this will include your full peer review and any attached files.

Reviewer #1: **Yes: **Nicholas E Pingitore

Reviewer #2: No

Reviewer #3: No

---

## [Editor Report · Decision Letter 1]

14 Jan 2022

Archaeometric studies on rock art at four sites in the northeastern Great Basin of North America

PONE-D-21-23506R1

Dear Dr. Andreae,

We’re pleased to inform you that your manuscript has been judged scientifically suitable for publication and will be formally accepted for publication once it meets all outstanding technical requirements.

Kind regards,

Fabio Marzaioli, Ph.D

Academic Editor

PLOS ONE
---

## [Editor Report · Acceptance letter]

17 Jan 2022

PONE-D-21-23506R1 

Archaeometric studies on rock art at four sites in the northeastern Great Basin of North America 

Dear Dr. Andreae:

I'm pleased to inform you that your manuscript has been deemed suitable for publication in PLOS ONE. Congratulations! Your manuscript is now with our production department. 

Kind regards, 

on behalf of

Dr. Fabio Marzaioli 

Academic Editor

PLOS ONE